# Exploring the structural landscape of DNA maintenance proteins

Kenneth Bødkter Schou [1,2,3] ✉, Samuel Mandacaru[2], Muhammad Tahir[2], Nikola Tom[4], Ann-Sofie Nilsson[3], Jens S. Andersen [2], Matteo Tiberti [5], Elena Papaleo [5,6] & Jiri Bartek [1,3] ✉

Evolutionary annotation of genome maintenance (GM) proteins has conventionally been established by remote relationships within protein sequence databases. However, often no significant relationship can be established. Highly sensitive approaches to attain remote homologies based on iterative profile-to-profile methods have been developed. Still, these methods have not been systematically applied in the evolutionary annotation of GM proteins. Here, by applying profile-to-profile models, we systematically survey the repertoire of GM proteins from bacteria to man. We identify multiple GM protein candidates and annotate domains in numerous established GM proteins, among other PARP, OB-fold, Macro, TUDOR, SAP, BRCT, KU, MYB (SANT), and nuclease domains. We experimentally validate OB-fold and MIS18 (Yippee) domains in SPIDR and FAM72 protein families, respectively. Our results indicate that, surprisingly, despite the immense interest and long-term research efforts, the repertoire of genome stability caretakers is still not fully appreciated.

Genomes of all cellular lifeforms are plagued by the threat of DNA damaging insults, mutations, or copying errors. To counteract the potentially deleterious consequences of such insults, organisms have evolved systems that safeguard genetic information. Past studies have unearthed a plethora of proteins categorized structurally and functionally into several independent DNA repair networks. In humans alone, some >500 caretakers are directly or indirectly involved in GM, and almost as many are engaged in the mitotic and chromosome segregation processes. DNA damage response (DDR) proteins, however, are constructed from combinations of much fewer evolutionary conserved modules, that is by shuffling and recombining a limited repertoire of conserved domain precursors. Hence, the identification and categorization of proteins bearing such conserved structural entities can provide tangible insight into hitherto unknown protein functions. Previous evolutionary annotation methods have been highly successful in uncovering unknown relationships between DNA repair systems[1,2]. However, such profile-to-sequence comparative analysis often fails to identify any significant homology, revealing a limitation of this approach. This is certainly the case for the most intriguing group of DDR proteins in humans, namely those for which no orthologs have yet been studied. Both sensitivity and specificity of the homology searches can be dramatically improved by comparing sequence profiles through iterative profile-to-profile algorithms such as the hidden Markov model (HMM)-based iterative profile-HMM searches[3,4]. These methods compare the profiles of both query and target by exploiting databases of HMMs (such as the protein family (PFAM) database) in which protein profile HMMs rather than sequences are compiled. Profile HMMs are superior to simple sequence

[1]Genome Integrity, Danish Cancer Institute, Danish Cancer Society, Strandboulevarden 49, 2100 Copenhagen, Denmark. [2]Department of Biochemistry and Molecular Biology, University of Southern Denmark, Campusvej 55, 5230 Odense M, Denmark. [3]Division of Genome Biology, Department of Medical Biochemistry and Biophysics, Science for Laboratory, Karolinska Institute, Solna 171 77, Sweden. [4]Lipidomics Core Facility, Danish Cancer Institute (DCI), DK-2100 Copenhagen, Denmark. [5]Cancer Structural Biology, Danish Cancer Society Research Center, Strandboulevarden 49, 2100 Copenhagen, Denmark. [6]Cancer Systems Biology, Section for Bioinformatics, Department of Health and Technology, Technical University of Denmark, 2800 Lyngby, Denmark. ✉e-mail: kensch@cancer.dk; jiri.bartek.1@ki.se

profiles since in addition to the amino acid frequencies identified in a multiple sequence alignment, they include the position-specific probabilities for inserts and deletions along the alignment. Among these, profile-HMMs have emerged as powerful tools in decoding the structural and functional landscape of genome maintenance proteins. For example, the elucidation of the S,T-Q phosphopeptide-binding BRCT domain, initially discovered in breast cancer susceptibility protein BRCA1[5] and later identified in many other proteins almost exclusively functioning in DNA damage response pathways[6], has been greatly facilitated by the advent of profile-to-sequence and later profile-HMM- based computational database surveys, enabling reliable detection of subtle sequence homologies indicative of shared structural motifs[7–10]. Similarly, the OB fold domain, known for its role in nucleic acid binding and recognition, has recently witnessed a surge in profile-HMM applications[11–17]. Profile-HMMs, with their ability to capture remote homologies, have proven indispensable in accurately identifying OB-fold-containing proteins, offering valuable insights into their evolutionary relationships and functional implications. Previously, the systematic application of profile-HMMs has emerged as an efficient tool for identifying DNA repair protein structures in bacteria and metazoa. These studies individually surveyed nuclease and OB fold-bearing protein sequences, effectively revealing family members within DNA repair pathways[18–20].

By leveraging the evolutionary information encoded in sequences, profile-HMMs have played a pivotal role in advancing our understanding of DNA repair mechanisms, offering a precise and efficient method for the computational annotation of protein structures associated with this vital cellular process. Although profile-HMM methods have been in existence for nearly two decades[21], it is only in recent times that their complete potential, robustness, and accuracy in large-scale protein structural assessment through the AlphaFold methods have become evident[22]. Other recent methods for structural prediction and biomolecule interactions using deep learning, such as convolutional and graph neural networks[23,24], have been developed. However, the latest version of AlphaFold, AlphaFold 3[25], allowing for the highest degree of precision achieved so far in predicting protein structures and protein-biomolecule interactions. To our knowledge, however, the repertoire of GM proteins has so far not been the subject of systematic comparative analysis across domains and species using these state-of-the-art computational profile-HMM strategies. Hence, given the efficacy of the computational methods in dissecting the protein structures and functions, we resorted to an in-depth sequence analysis of all known and putative players in the DNA maintenance networks. Here, we report the results of such an analysis and discuss several previously undetected conserved domains that we have uncovered in the present study.

## Results
### Overview of the genome maintenance architectural landscape analyses
A key goal of comparative sequence studies is the annotation of conserved domains as well as the discovery of structural and evolutionary relationships between cataloged domains. To understand the variety of GM protein structures across eukaryotes, we set out to answer two questions concerning GM architectures and their evolution. First, given the adeptness of profile-HMM methods in detecting remote homologies, can they be applied systematically to uncover unknown structures and relationships of in the human GM proteome? Second, despite the seemingly past eminent functional and evolutionary characterization of existing GM proteins, can we identify additional players in the human DNA damage responses? We, therefore, examined the structural characteristics and relationships among GM proteins across species by methodically evaluating their sequence homologies with protein profiles in the PFAM database through sensitive hidden Markov-based profile-to-profile (profile-HMM) searches.

This information was then used to explore possible remote and undiscovered relationships within the human proteome. The result of this survey is summarized in Fig. 1 and -supplementary Fig. 1 and discussed in additional detail in the following sections. Collectively, we uncovered 113 unknown evolutionary conserved protein families, 59 predicted structures in established GM proteins, and 54 structures in uncharacterized GM candidate proteins.

The sequences and structures of the main catalytic domains of many GM proteins such as polymerases, helicases, and other ATPases have been characterized in detail previously and are readily recognizable due to the conservation of diagnostic motifs. Consequently, the analysis did not considerably expand these protein superfamilies. Most protein modules identified in our survey belong to either DNA binding domain families or to protein adapters i.e., protein-protein binding interfaces. Characteristically, binding domains are imperfect and show much less sequence conservation than enzymes, which is likely why many binding domains have remained undetected up to this point.

### Adapter domain discovery
Noteworthy examples of the previously unknown (by the time of this analysis) DNA binding domain family members are seven MYB (SANT) domain proteins, nine TUDOR domain proteins, six OB-fold proteins, four Mis18 (yippee) domain proteins, 19 SAP domain proteins, three WSD domain proteins, and two KU70 or KU86 beta-barrel domain proteins (Fig. 1b and Supplementary Fig. 1a, b). These domains occur in both well-described GM proteins as well as in uncharacterized proteins (Supplementary Fig. 1a, b). Among established GM proteins and candidates, we identified adapter or DNA-binding domains in 62 proteins. Examples of these are four UBA ubiquitin-binding, five NNCH, three SFI1, two BRCT (Supplementary Data 1), two BAH, and one POLO box domains (Fig. 1b–p and Supplementary Fig. 1a, b). We also uncovered members of six protein families tied to mitotic functions. (Fig. 1l–p and Supplementary Fig. 2d). Four of the latter families have kinetochore ontologies, i.e., the protein members are enriched at kinetochore complexes required for microtubule attachment to centromeres during mitotic chromosomal segregation (Fig. 1l–p and Supplementary Fig. 2d). We also expand our previous discovery of the kinetochore NDC80 (NUF) calponin (NNCH) subfamily domains[26] by identifying five additional members: CEP44, HAUS3, HAUS6, HAUS7, and TEDC1 (Fig. 1o and Supplementary Fig. 2a–c). As with other members of the NNCH family, the N-terminal NNCC domain is adjoined to a C-terminal region of disparate arrays of heptad repeats (coiled-coils) (Supplementary Fig. 2a) as predicted by weighted and unweighted matrices[27] indicating that these proteins are members of an evolutionary conserved family of bimodular coiled-coil proteins.

The majority of DNA repair proteins bear catalytic domains and DNA binding domains (Fig. 1g). A noteworthy exception to this tendency is the multiprotein Integrator (Int) complex involved in the promotion of DNA repair and G2 to M checkpoint through crosstalk with the multiprotein complex sensor involved in sensing ssDNA (SOSS) and in small nuclear RNAs (snRNA) transmission[28]. In addition, our analysis extends the structural features of these Integrator complex subunits including the INTS2 and INTS10 that currently have no known conserved structures. We find that INTS2 and INTS10 are comprised almost entirely of disparate arrays of TPRs units (Supplementary Fig. 3a–c). We collectively designate these alpha-helical repeats in the Int proteins alpha solenoid repeats (Supplementary Fig. 3a). Interestingly, our analysis also revealed that the Int complex member INTS14 is a thus far unknown paralog of the NHEJ repair proteins KU70 (KU86). INTS14 bears, besides an N-terminal vWA Von Willebrand factor type A domain - as do the INTS6 and INTS13 subunits (Supplementary Fig. 3a) - a predicted high confidence KU_core domain adjoined to a KU_C C-terminal domain (Supplementary Fig. 3a, d). Supporting our structural prediction of INTS14 by sequence, recently

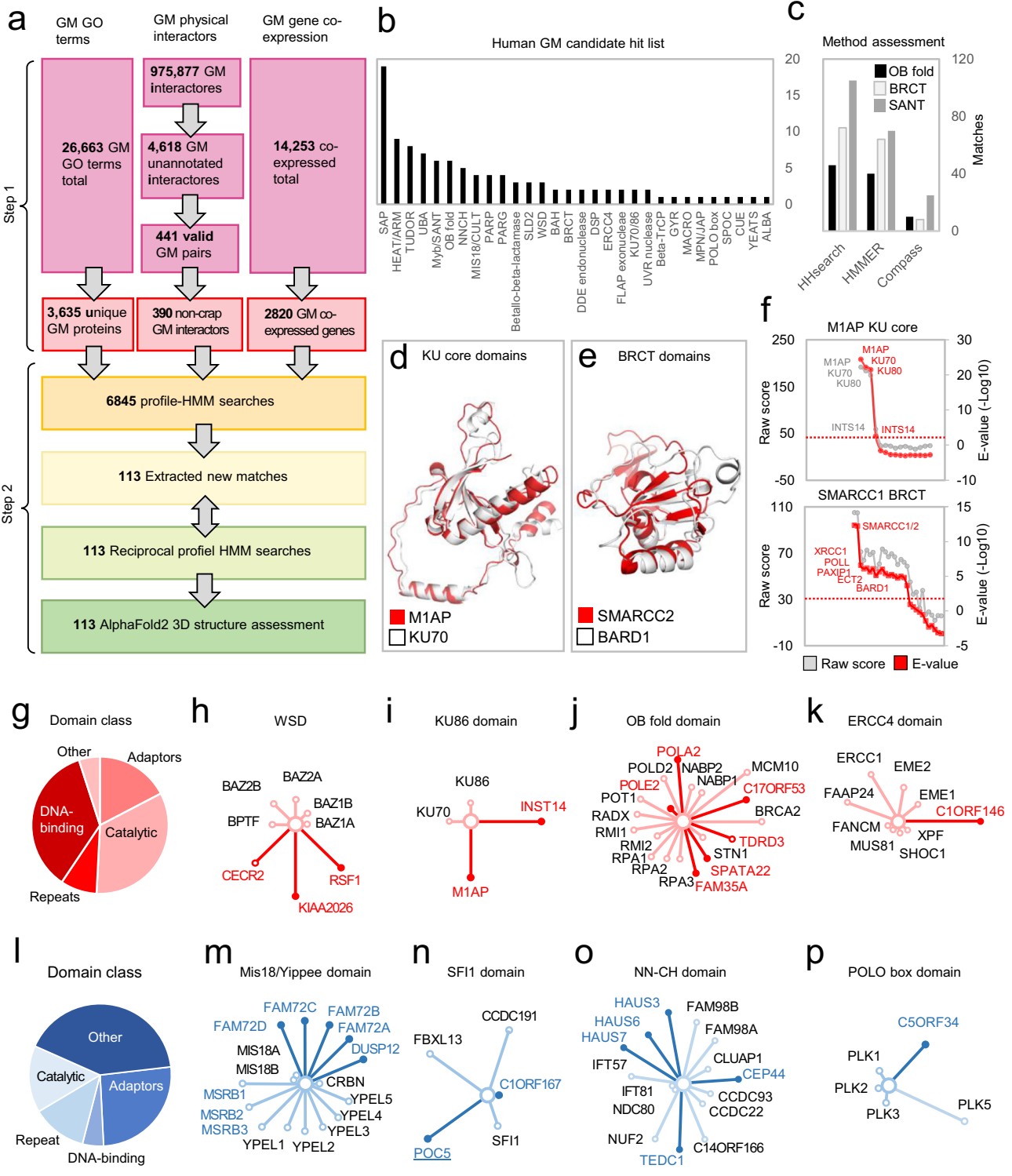

using crystallography the structure of INTS14 in complex with INTS13 was shown to adopt a KU70 (KU86) complex-like structure[29]. Surprisingly, this overall architecture resembling the KU70 (KU86) proteins was also identified for the M1AP protein (Fig. 1d, f, Supplementary Fig. 3a, d, and Supplementary Data 3), suggesting that humans bear four KU paralogs (Fig. 1i). The M1AP structural homology to the Ku70-Ku86 DNA repair complex proteins suggests nucleic acid affinity in DNA repair. M1AP was previously shown to be almost exclusively expressed in testis and having functions in male germ cell development and in meiosis[30] indicating that M1AP might be implicated in meiotic

recombination events. Indeed, an *M1AP* gene co-expression analysis of testis mRNA using the GTEx RNA seq data showed that M1AP is strongly co-regulated with a network of DNA repair and DDR-related genes (Supplementary Fig. 3e, f), suggesting that M1AP may be involved in DNA repair processes.

**Predicted family members of the poly(ADP-ribose)-related enzymatic processes**
While catalytic domains are generally well conserved, making it less likely to identify proteins in this category, we have now nevertheless

**Fig. 1 | Overview of the computational survey and summary of results. a**, In step 1 unique GM proteins were compiled by three different approaches. Flow diagram summarizing data collection and subsequent sequence searches. GO terms for GM proteins were compiled in the Amigo database[105] using four search terms across species yielding a total of 28,663 GO terms. Of these, 3635 are unique GM proteins from the seven selected organisms namely *H. sapiens*, *D. melanogaster*, *C. elegans*, *A. thaliana*, *S. cerevisiae*, *S. pombe*, and *E. coli* (K12). GM physical interactors were retrieved from the IID database yielding a total of 975,877 interactors across species. Among these are 4618 interactors not previously implicated in GM. Of these, only 441 interactor pairs include one established DNA repair protein and one protein not previously linked to the DDR. Among these, 51 interactors were identified as recurrent contaminants in the CRAPome database yielding a final list of 390 unique GM interactors not previously implicated in the DDR. GM gene co-expressed genes were retrieved from the CEMiTool identifying a total of 36,410 GM co-expressed genes. Among these CEMiTool identifies 3523 overlapping co-expressed genes between two different tissues, which upon filtering for house-keeping genes and registered Crapome entries are reduced to 2820 co-expressed gene pairs (of which one gene per pair is an established GM gene). In Step 2 the compiled list of 4395 unique GM proteins were used as search queries in profile-HMM searches. These searches yielded a total of 108 hitherto unknown human domains in established and candidate GM proteins. These were used as seeds for reciprocal profile-HMM searches resulting in 108 validated candidate domains. Finally, the valid candidates were assessed by AlphaFold2 3D structural modeling to

structurally validate the predicted evolutionary relationships across protein families. **b** Summary of identified classes of protein domains in the human proteome. **c** Validation of profile-HMM methods. Three methods were tested for their efficiency in detecting homologous protein domains in the protein databank (PDB). The three protein domains used as seeds were human RPA1 OB_2, the human BARD1 BRCT, and human MYB (SANT) domains. **d, e** Examples of predicted 3D structures of two identified candidate domains as judged by AlphaFold modeling. Predicted domains were superimposed with closest paralog domains in Pymol as indicated. **f** Probability plots of profile-HMM remote homology searches using either the predicted KU core domain of M1AP or the predicted BRCT domain of SMARCC1 as sequence queries. **g–k** Summary of examples of DNA repair candidates identified in the computational survey shown in red. **g** Summarizes DNA repair protein domain classes. **h–k** Four examples of identified DNA repair candidates as judged by their predicted protein domains. Red nodes represent candidates (at the time of analysis). The length of nodes from the center corresponds to the sequence homology of the signature domain family profile. **l–p** Summary of mitotic candidates identified in the computational survey shown in blue. **l** Summarizes mitotic protein domain classes. **m–p** Four examples of identified mitotic candidates as judged by their predicted protein domains. Blue nodes represent candidates (at the time of analysis). The length of nodes from the center corresponds to the sequence homology of the signature domain family profile. Source data are provided as a Source Data file.

uncovered several human enzymes, including six nucleases, three poly(ADP-ribose) polymerases (PARPs), four Poly (ADP-ribose) glyco-hydrolase (PARG)-like proteins, and three metallo-beta-lactamases (Figs. 1b, 1g–k and Supplementary Fig.1a, b). Because the gene products of two of the identified PARPs, namely TEX15 and TASOR, are widely expressed putative GM players in thus far meiotic recombination or chromatin remodeling processes[31,32], respectively, and their PARP domains share overall sequence and predicted structure (Fig. 2d, f), this raises the possibility that TEX15 and TASOR, as well as the uncharacterized TASOR2 (FAM208B), could function as PARPs in the DDR analogous to e.g., PARP-1. Interestingly, recently using cryo-electron microscopy, a cryptic PARP domain was identified in TASOR. This domain is catalytically inactive and dispensable for assembly and chromatin targeting but is critical for epigenetic regulation of target histone H3K9me3[33]. This PARP domain corresponds to the first PARP domain in TASOR uncovered in our sequence analysis, whereas the second PARP domain that we predict is unknown (Fig. 2c). Since TASOR and TASOR2 are paralogs and share high overall sequence homology, we assessed whether TASOR2 might share PARP domain properties with the cryptic PARP domain of TASOR. Indeed, an inspection of the primary structure of the TASOR2 PARP domain indicated that several residues essential for the catalytic function, are lost in the PARP domain of TASOR2 as with TASOR Fig. 2a, b. Hence, the PARP domains of TASOR and TASOR2 might resemble the cryptic catalytic dead PARP domain of PARP13, as previously shown for TASOR. In support of this notion, the 3D model of TASOR2 predicted by ColabFold indicated that its PARP domain adopts a closed structural loop comparable to the PARP13 catalytic domain Fig. 2e.

The four hitherto unknown PARG domains identified with high confidence (Fig. 2f) in the putative A-kinase anchor proteins AKAP3, AKAP4, SPHKAP, and AKAP11, all show high sequence conservation and predicted 3D structure to the C-terminal half of the PARG catalytic domain (Fig. 2g, h and Supplementary Fig. 6a). The remainder N-terminal portion in these AKAPs appears to have been lost during evolution. Because the predicted PARG domain residues in these AKAPs show high conservation compared to residues in PARG that are configured in close contact with its bound poly(ADP-ribose) (PAR) moiety (Supplementary Fig. 6b), it is conceivable that the AKAP's PARG domain evolved specifically to bind PAR-modified proteins. Indeed, a closer examination of the PARG homologies between identified AKAPs (and the PARG catalytic domain) revealed that this portion corresponds to the PARG macro domain (Fig. 2g).

## DNA-binding domains

The class of protein domains of which we identify most candidates is the DNA-binding domain. Three categories of DNA-binding domains are overrepresented among the candidates. These are the MYB (SANT), SAP, and OB-fold domains (Fig. 1b and Supplementary Fig.4). Our analysis expands the superfamily of MYB (SANT) domain-containing proteins with six members, namely TIMELESS, SMARCC1, SMARCC2, CRAMP1, LOC100506514, and LOC107985532 (Supplementary Fig. 4a, b). The outermost C-terminal region comprising the previously unknown MYB (SANT) domains in TIMELESS (Supplementary Fig. 4a, d) is required for proficient circadian clock rhythm regulation in fruit flies[33], suggesting an important biological function of these two MYB (SANT) domains in TIMELESS.

We also predict 19 SAP candidate domains that all appear to adopt similar 3D structures (Supplementary Fig. 4e). Several of the predicted SAP domains were previously annotated as distinct domain classes such as the LEM domains in e.g., LETMD1, LETM2, LETM1, the ARMET domain in MANF and CDNF (Supplementary Fig. 4e), as well as the NCD1 domain in NAB1 and NAB2. These domains were previously shown to individually adopt a three alpha-helix topology characteristic of the SAP domain[34], but their mutual evolutionary relationship to the SAP domain was unknown. Our remote homology searches also revealed an until now unknown SAP domain in human PARP1 (233-287) (Fig. 2c). Previously, the region encompassing the predicted SAP domain was designated the human *PARP1 domain C*[35,36]. Further inspection of domain C indicated, however, that it might consist of two subdomains i.e., a SAP domain comprising the N-terminal half and a C-terminal zinc ribbon fold comprising the third zinc-binding domain of PARP1[37].

Among the DNA-binding domains, the Oligonucleotide or Oligosaccharide-binding (OB)-fold domain is one of the most common domains in DNA repair proteins[38]. We identified previously unde-tected, distinct versions of the OB-fold domain in the families of repair proteins (Fig. 1b, j and Supplementary Fig. 1a, b). In each case, the amino acid signature typical of OB-fold domains is modified and not easily recognizable, but all show a remote relationship to the OB-fold domains seen in bacterial proteins. Significant similarity to OB-fold domains can be demonstrated for these domains for all known DNA-binding OB-fold-containing proteins as well as in six human proteins (at the time of this analysis), namely SHLD2 (RINN2 or FAM35A), HROB (C17ORF53), SPIDR, POLE2, SPATA22, and TDRD3 (Fig. 1j and Supplementary Fig. 1 and Supplementary Data 2). Structural characterization of the OB-fold domains and their role in the DDR of human SHLD2,

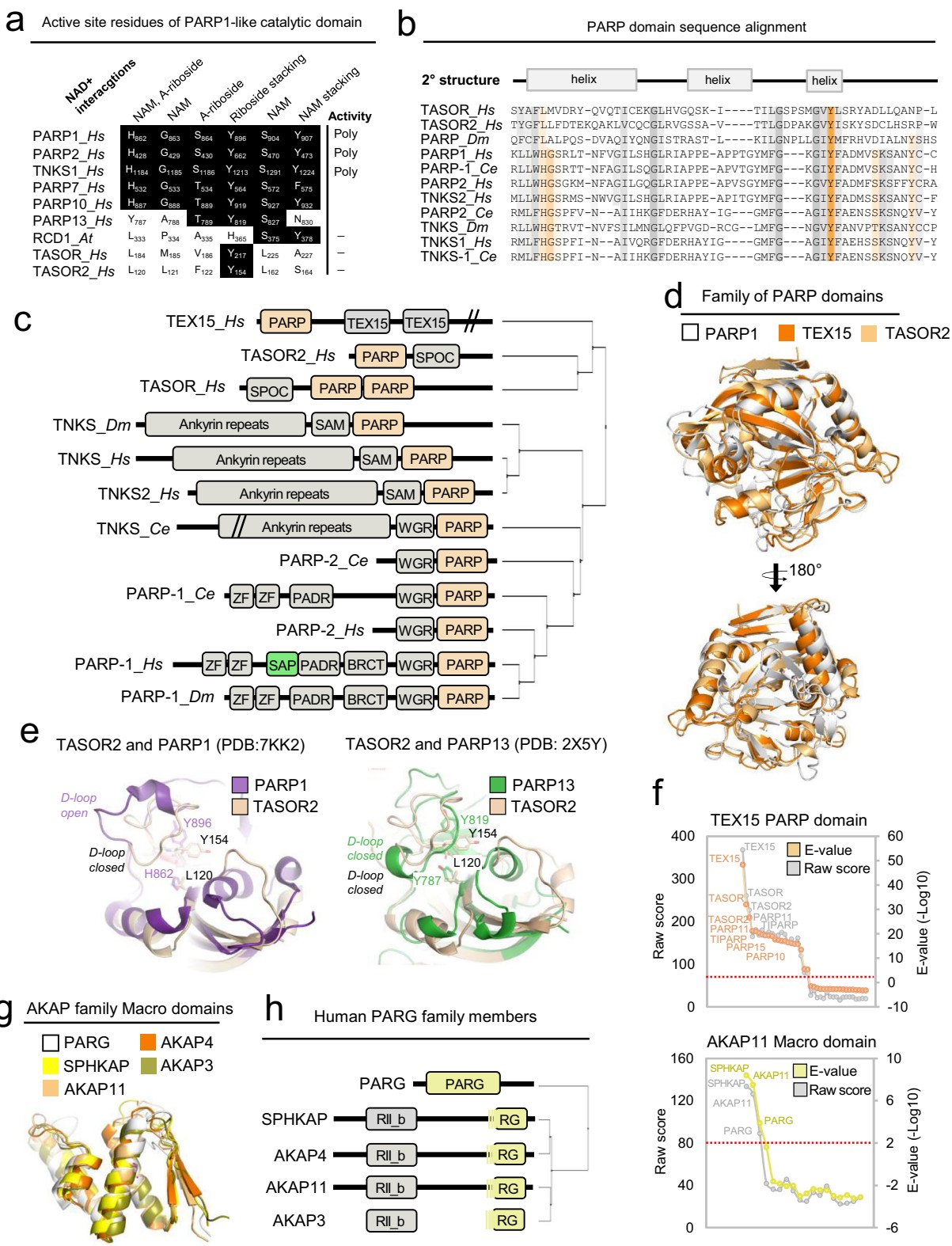

**a** Active site residues of PARP1-like catalytic domain

| NAD+ interacgtions | NAM, A-riboside | NAM | A-riboside | Riboside stacking | NAM | NAM stacking | Activity |
|---|---|---|---|---|---|---|---|
| PARP1_*Hs* | $H_{862}$ | $G_{863}$ | $S_{864}$ | $Y_{896}$ | $S_{904}$ | $Y_{907}$ | Poly |
| PARP2_*Hs* | $H_{428}$ | $G_{429}$ | $S_{430}$ | $Y_{662}$ | $S_{470}$ | $Y_{473}$ | Poly |
| TNKS1_*Hs* | $H_{1184}$ | $G_{1185}$ | $S_{1186}$ | $Y_{1213}$ | $S_{1291}$ | $Y_{1224}$ | Poly |
| PARP7_*Hs* | $H_{532}$ | $G_{533}$ | $T_{534}$ | $Y_{564}$ | $S_{572}$ | $F_{575}$ | |
| PARP10_*Hs* | $H_{887}$ | $G_{888}$ | $T_{889}$ | $Y_{919}$ | $S_{927}$ | $Y_{932}$ | |
| PARP13_*Hs* | $Y_{787}$ | $A_{788}$ | $T_{789}$ | $Y_{819}$ | $S_{827}$ | $N_{830}$ | – |
| RCD1_*At* | $L_{333}$ | $P_{334}$ | $A_{335}$ | $H_{365}$ | $S_{375}$ | $Y_{378}$ | – |
| TASOR_*Hs* | $L_{184}$ | $M_{185}$ | $V_{186}$ | $Y_{217}$ | $L_{225}$ | $A_{227}$ | – |
| TASOR2_*Hs* | $L_{120}$ | $L_{121}$ | $F_{122}$ | $Y_{154}$ | $L_{162}$ | $S_{164}$ | – |

**b** PARP domain sequence alignment

2° structure — helix — helix — helix

```
TASOR_Hs    SYAFLMVDRY-QVQTICEKGLHVGQSK-I----TILGSPSMGVYLSRYADLLQANP-L
TASOR2_Hs   TYGFLLFDTEKQAKLVCQCGLRVGSSA-V----TTLGDPAKGVYISKYSDCLHSRP-W
PARP_Dm     QFCFLALPQS-DVAQIYQNGISTRAST-L----KILGNPLLGIYMFRHVDIALNYSHS
PARP1_Hs    KLLWHGSRLT-NFVGILSHGLRIAPPE-APPTGYMFC---KGIYFADMVSKSANYC-C
PARP-1_Ce   RLLWHGSRTT-NFAGILSQGLRIAPPE-APVTGYMFG---KGIYFADMVSKSANYC-H
PARP2_Hs    RLLWHGSGKM-NFAGILGQGLRIAPPE-APVSGYMFG---KGVYFADMFSKSFFYCRA
TNKS2_Hs    MLLWHGSRMS-NWVGILSHGLRIAPPE-APITGYMFG---KGIYFADMSKSANYC-F
PARP2_Ce    RMLFHGSPFV-N--AIIHKGFDERHAY-IG---GMFG---AGIYFAENSSKSNQYV-Y
TNKS_Dm     RLLWHGTRVT-NVFSILMNGLQFPVGD-RCG--LMFG---NGVYFANVPTKSANYCCP
TNKS1_Hs    RLLFHGSPFI-N--AIVQRGFDERHAYIG----GMFG---AGIYFAEHSSKSNQYV-Y
TNKS-1_Ce   RMLFHGSPFI-N--AIIHKGFDERHAYIG----GMFG---AGIYFAENSSKSNQYV-Y
```

**c**

TEX15_*Hs* — PARP — TEX15 — TEX15 //
TASOR2_*Hs* — PARP — SPOC
TASOR_*Hs* — SPOC — PARP — PARP
TNKS_*Dm* — Ankyrin repeats — SAM — PARP
TNKS_*Hs* — Ankyrin repeats — SAM — PARP
TNKS2_*Hs* — Ankyrin repeats — SAM — PARP
TNKS_*Ce* — Ankyrin repeats — WGR — PARP
PARP-2_*Ce* — WGR — PARP
PARP-1_*Ce* — ZF — ZF — PADR — WGR — PARP
PARP-2_*Hs* — WGR — PARP
PARP-1_*Hs* — ZF — ZF — SAP — PADR — BRCT — WGR — PARP
PARP-1_*Dm* — ZF — ZF — PADR — BRCT — WGR — PARP

**d** Family of PARP domains

☐ PARP1   ■ TEX15   ■ TASOR2

180°

**e**

TASOR2 and PARP1 (PDB:7KK2)
■ PARP1   ■ TASOR2
*D-loop open*   Y896   Y154   *D-loop closed*   H862   L120

TASOR2 and PARP13 (PDB: 2X5Y)
■ PARP13   ■ TASOR2
Y819   Y154   *D-loop closed*   *D-loop closed*   Y787   L120

**f**

TEX15 PARP domain
(Raw score / E-value (-Log10))
E-value (orange), Raw score (grey)
TEX15, TASOR, TASOR2, PARP11, TIPARP, PARP15, PARP10

AKAP11 Macro domain
(Raw score / E-value (-Log10))
E-value (yellow), Raw score (grey)
SPHKAP, AKAP11, PARG

**g** AKAP family Macro domains

☐ PARG   ■ AKAP4
■ SPHKAP   ■ AKAP3
■ AKAP11

**h** Human PARG family members

PARG — PARG
SPHKAP — RII_b — RG
AKAP4 — RII_b — RG
AKAP11 — RII_b — RG
AKAP3 — RII_b — RG

HROB, SPIDR, and SPATA22 was recently reported[15,16,39–46], whereas SPIDR and TDRD3 remain structurally uncharacterized. Recently, the SPIDR protein was identified in a DDR protein complex with BLM helicase that functions in the repairing of DNA double-strand breaks (DSBs) through the non-homologous end joining (NHEJ) pathway[47]. While the structure of SPIDR and its DNA-binding properties are currently unknown, our sequence analysis predicts three OB folds in the SPIDR C-terminus with high confidence (Fig. 3a–e). These OB folds appeared to adopt a C-terminal tandem arrangement including a zinc-ribbon in the outermost C-terminal OB-fold previously shown for other DNA repair proteins such as RPA1, RADX, MEIOB, and SHLD2[38]. Subsequent searches using the overall structure comprising these three SPIDR OB folds as a seed against protein tertiary structure databases using Foldseek[48] matched the three C-terminal OB-folds in *S. cerevisiae*

**Fig. 2 | Identified Poly-(ADP-Ribose) catalyzing protein families. a** Diagram of the active site residues of the PARP1-like catalytic domain. Similar to PARP13, TASOR and TASOR2 have lost essential residues required for catalytic activity. **b** MSA of selected PARP domain sequences including those of TEX15, TASOR, and TASOR2. Conserved residues are shown in orange as assessed by Clustal W with modifications. Predicted secondary structures are shown above the MSA. Boxes indicate alpha-helices, and arrows indicate beta-sheets. **c** Schematic domain architectures of selected human PARPs including the three PARP candidates TEX15, TASOR, and TASOR2. Phylogenetic trees were calculated from MSA average distances using approximately the maximum-likelihood method in the IQ-Tree v.2.050 program. **d** Predicted PARP domains of TEX15 and TASOR2 as assessed by AlphaFold. PARP domains of TEX15 and TASOR2 (orange) were superimposed with the PARP domain of PARP1 (white) in Pymol. **e** Superimposed PARP domains of either TASOR2 and PARP1 (left) or TASOR2 and PARP13 (right). Conserved residues are indicated. **f** Probability plots of profile-HMM searches using the predicted TEX15 PARP domains (top) or the predicted AKAP11 Macro domain (bottom) as queries. **g** 3D structures of AKAP family member Macro domains as predicted by AlphaFold. AKAP Macro domains (yellow/orange nuances) are superimposed with the Macro domain in human PARG (white). **h** PARG domain relationships to human AKAP family proteins. The C-termini of SPHKAP, AKAP3, AKAP4, and AKAP11 show remote homology to the C-terminal portion of PARG comprising the PAR-binding Macro domain. Source data are provided as a Source Data file.

RPA1 as the top-ranking significant match (Fig. 3d). AlphaFold2 structural modeling of SPIDR full-length structure predicted a fourth central globular domain adjoined to the N-terminal disordered region (Fig. 3e). Our remote homology sequence searches detect this domain with only low significance (E = 0.015) and hence the nature of this domain remains inconclusive. We therefore cautiously designate this fourth domain an inferred OB-fold (Fig. 3e). Interestingly, we noticed that a subset of the OB-fold-bearing proteins shows striking similar overall structural architecture i.e., they bear an N-terminal intrinsically disordered portion adjoined to a C-terminus bearing one or more OB-fold domains (Fig. 3e), indicating that these proteins might be evolutionarily and functionally closely related. This organization is reminiscent of the SHLD2 (RINN2), HROB, and SPATA22 all bind ssDNA through their OB-fold domains leading us to hypothesize that SPIDR may bind ssDNA. To test this notion, we tested a FLAG-tagged version of SPIDR for its ability to bind Biotin-labeled double and single-stranded DNA (dsDNA or ssDNA). Indeed, streptavidin pulldown assays of Biotin-ssDNA avidly retrieved FLAG-SPIDR expressed in HEK293T cells whereas FLAG-SPIDR showed much reduced binding efficiency to Biotin-dsDNA (Fig. 3h). The OB-fold region of SPIDR is responsible for the binding to DNA as full-length FLAG-SPIDR and a FLAG-3XOB fragment but not FLAG-SPIDR lacking its OB-fold domains could be pulled down with Biotin-ssDNA (Fig. 3e, f). Furthermore, based on alignment with disruptive mutations in the RPA1 DBD-C OB fold (C505A and C508A) that result in a marked reduction in ssDNA-binding affinity[49], specific point mutations within the SPIDR OB3 domain (C817A and C820A) reduced its interaction with ssDNA markedly (Fig. 3i, j). In addition, using the recently developed Alpha-Fold 3 method for modeling protein binding to diverse biomolecules, we assessed the predicted propensity of the OB3 domain to bind DNA. Interestingly, when using a random DNA sequence as a query, a stretch of ssDNA, but not the corresponding RNA sequence, was predicted to bind the SPIDR OB3 domain with high significance (Fig. 3k), further supporting the notion that the OB-folds of SPIDR bind ssDNA. Recently causal polymorphisms in SPIDR were implicated in primary ovarian insufficiency (POI) with patient cells showing chromosomal instability[50,51]. Both pathogenic SPIDR variants are nonsense mutations (R272*, W280*) and lie near the inferred OB-folds (Fig. 3e). Hence these variants are predicted to delete most of the OB-fold containing portion in SPIDR. To test whether such disease truncations will affect the ability of SPIDR to bind to chromatin, we expressed FLAG-tagged full length, the W280*-fragment (SPIDR 1-280), or a C-terminal portion only containing OB folds (623-915) of SPIDR in U2OS cells and isolated chromatin fractions. Indeed, Full-length SPIDR and the OB-fold C-terminus but not the W280* fragment associated with chromatin (Fig. 3g), suggesting that the nonsense mutations seen in SPIDR in POI patients disrupt the SPIDER protein's DNA-binding capabilities.

## The FAM72 family proteins are MIS18 (Yippee) paralogs that oligomerize and bind RPA proteins

We identified the DNA binding domain MIS18 (yippee) in four human FAM72A paralogs (FAM72A-D) (Fig. 4a–c). The MIS18 (yippee) domain is found in the kinetochore proteins MIS18a, MIS18b, and yeast homolog MIS18[52] as well as the poorly understood Yippee zinc-binding and DNA-binding proteins YPEL1-5. Using AlphaFold2, we found that the predicted tertiary structure of FAM72A-D resembles the solved 3D structure of human MIS18A (PDB: 5HJ0), indicating that the FAM72A-D homologies by sequence to the MIS18 (Yippee) family are genuine (Fig. 4d). Because all four FAM72A paralogs are predicted to adopt almost the same 3D structures (Fig. 4d) and display high protein sequence identity (≥98%), this raises the possibility that they all function in the same molecular pathway(s) in the cell. Indeed, a network deconvolution analysis of RNA-seq data derived from a broad array of human tissues[53] to define the transcriptional signatures of *FAM72A-D* co-expressed genes revealed that all four paralogs share highly similar gene co-expression signatures (Fig. 4f and Supplementary Fig. 5c). Interestingly, among the top co-expressed genes across FAM72 family members are the *FAM72A-D* genes themselves (Fig. 4g), suggesting that FAM72A-D proteins are tightly co-regulated. The Yippee-like domain of *S. pombe* MIS18 was recently shown to possess an intrinsic propensity to oligomerize[54], raising the possibility that the *FAM72A-D* gene co-regulation reflects their mutual propensity to oligomerize. To test this possibility, we used the AlphaFold multimer suite[55] to assess whether FAM72 species oligomerize. Indeed, this analysis predicted that FAM72A can homodimerize and heterodimerize with all other FAM72 species (Fig. 4e). These mutual interactions between FAM72 family members were also supported by GFP pulldown assays using cell extracts of HEK293T cells co-expressing GFP-FAM72A with either FLAG-tagged FAM72B, FAM72C, or FAM72D (Supplementary Fig. 5b), indicating that all FAM72 family members might partake in multimer complexes in analogous molecular processes in the cell. Our *FAM72* family gene co-expression analysis also revealed that, besides the mutual FAM72 gene family co-expression profiles (Supplementary Fig. 5c), *FAM72* family genes are co-regulated with genes with genome maintenance ontologies such as DNA biosynthesis processes, DNA replication, and DNA repair as well as mitosis and meiosis (Fig. 4f and Supplementary Fig. 5c). FAM72A was recently shown to interact with and inhibit base excision repair (BER) uracil-DNA glycosylase UNG in antibody diversification[56–58]. Because we found that FAM72A's human paralogs are analogous in structure, form dimer or multimer complexes, and share transcriptional network, we asked whether other FAM72A paralogs might function in the same molecular processes as FAM72A? Indeed, as an example, our FLAG pulldown analysis of FLAG-FAM72B followed by mass spectrometry of FLAG immunocomplexes revealed that FAM72B was also found to bind avidly to UNG (Fig. 4h and Supplementary Fig. 5a) as previously shown for FAM72A, suggesting that these paralogs may participate in the same pathway. Surprisingly, these pulldown assays also revealed that RPA single-stranded DNA binding protein complex members (RPA1-3) were among the most abundant proteins to co-elute with FLAG-FAM72B (Fig. 4h). Subsequent FLAG pulldown of high salt cell extracts from HEK293T cells expressing FLAG-FAM72B and immunoblot supported the interaction with RPA1 (Fig. 4i). Hence, Fam72B, and by extension other FAM72 family members, besides binding to UNG, also

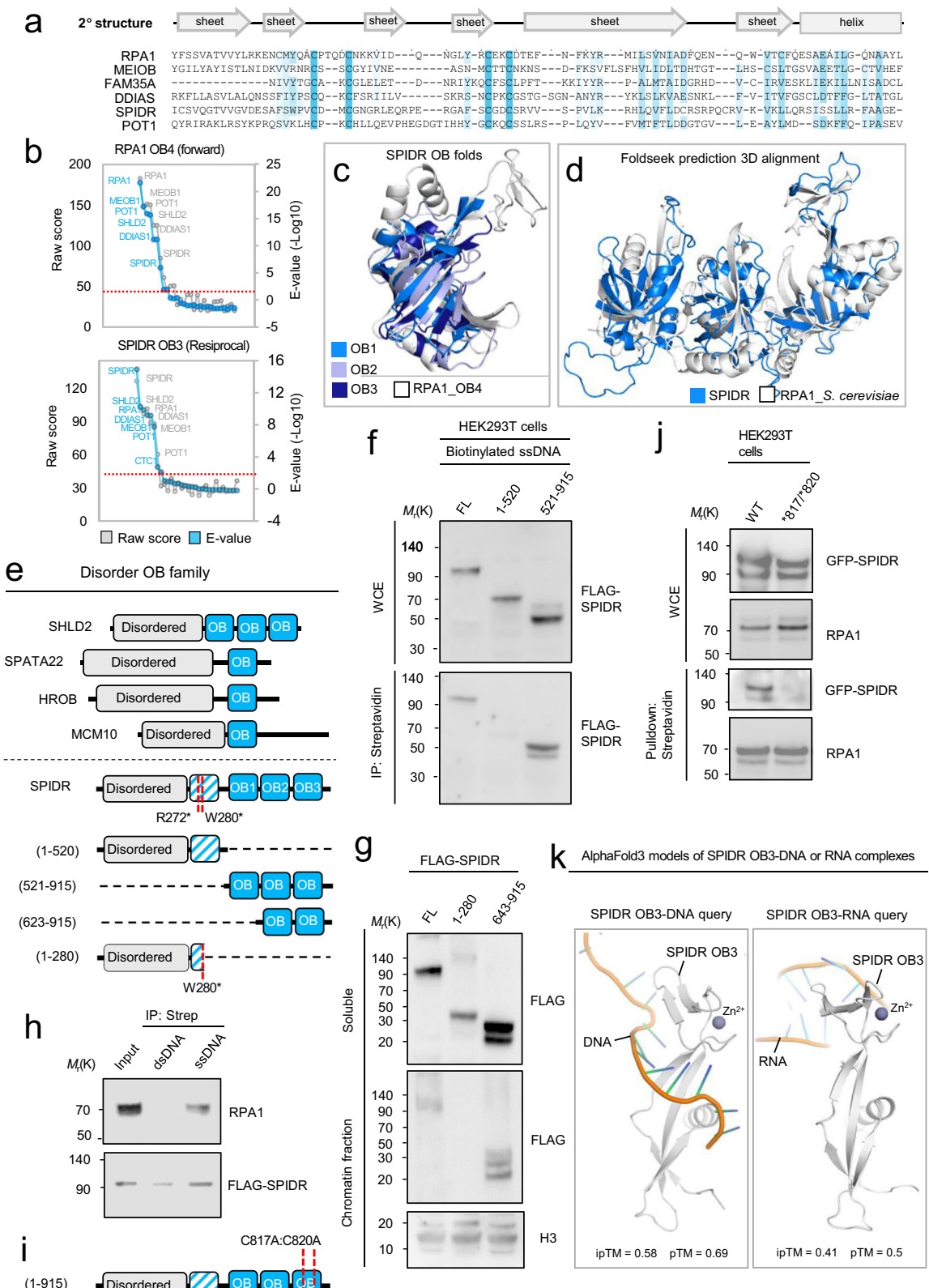

reside in complexes with RPA proteins, likely to function in DNA repair. To test this emerging possibility in more detail, we asked whether FAM72B might recruit to DNA during replication in the S phase as has been shown previously for RPA proteins[56,59]. Indeed, FLAG-FAM72B expressing cells released from a thymidine cell cycle block in late G1 showed gradually increased retention of FLAG-FAM72B on chromatin as cells traversed into the S phase as judged by immunoblotting of resolved proteins from the chromatin fraction (Fig. 4j). The chromatin accumulation of FAM72B in S phase coincides with RPA2 phosphorylation (Fig. 4j), supporting the notion that FAM72B and RPA proteins function in the same pathway. Because RPA proteins are well known to recruit to DNA upon various DNA insults due to the formation of naked ssDNA, we also tested if FAM72B accumulated on chromatin after short-term treatment with camptothecin (CPT), which

**Fig. 3 | Identification of OB fold domains in SPIDR. a** MSA of selected OB fold domain sequences including the outermost C-terminal OB fold of SPIDR. Conserved residues shown in blue were calculated using the Clustal W algorithm with modifications. Predicted secondary structures are shown above the MSA. Arrows indicate beta-sheets, and boxes indicate alpha-helices. **b** Probability plots of profile-HMM remote homology searches using either the human RPA1_OB4 domain (forward) or the predicted SPIDR OB3 domain (reciprocal) as sequence queries. **c** The three predicted OB folds in the SPIDR C-terminus. Here, AlphaFold predicted models are superimposed with the solved structure of RPA1_4OB fold DBD-D (PDB: 4GOP), shown in white. Short unstructured coils have been stripped off the SPIDR OB folds for clarity. **d** The three tandem OB-fold domains in the SPIDR C-terminus resemble that of other OB fold-containing proteins. Searching the overall predicted structure comprising these three SPIDR OB folds against protein structure databases using Foldseek (https://github.com/steineggerlab/foldseek) identifies S. cerevisiae RPA1 as the closest significant match. **e** Schematic illustration of the bimodular family of IDP and OB-fold family DNA repair proteins. Two causative

mutations identified in primary ovarian insufficiency (POI) patients are shown in SPIDR (red). **f** OB folds of SPIDR binds ssDNA. Cell extracts from HEK293T cells expressing indicated fragments of FLAG-SPIDR were incubated with biotinylated ssDNA and subjected to streptavidin pulldown using streptavidin resin. **g** Chromatin fractionation of HEK293T cells expressing either FLAG-SPIDR full length, truncated FLAG-SPIDR fragment corresponding to SPIDR containing disease mutation W280*. **h** FLAG-SPIDR binds ssDNA but not dsDNA. Cell extracts from HEK293T cells expressing FLAG-SPIDR were incubated with either biotinylated ssDNA or dsDNA and biotinylated DNA purified using a streptavidin (strep) resin. **i** Point mutations introduced into OB-fold domains of SPIDR. **j** Biotin-ssDNA pulldown analysis of cell extract from HEK293T cells expressing either GFP-SPIDR wildtype or a mutated version with the indicated amino acid substitutions. WCE = sample processing control. **k**, AlphaFold3 modeling of SPIDR OB3 domain in complex with either DNA or RNA. Immunoblots are representative results of two individual experiments (X = 2). Source data are provided as a Source Data file.

---

potently induces replication-dependent DSBs. FAM72B chromatin binding increases after CPT treatment (Fig. 4k), suggesting that FAM72B recruits to DNA following genotoxic insults in vertebrate cells. Finally, to further understand the functional relationship between FAM72 proteins and RPA, we examined the proficiency of RPA2 phosphorylation after replication stress in cells depleted for FAM72 proteins using FAM72 siRNA. Interestingly, FAM72 silencing reduced the phosphorylation of RPA2 after both 1.5 and 3 h of CPT exposure (Supplementary Fig. 5d), indicating that FAM72 proteins play an active role in the regulation of RPA following replication stress.

## Nuclease domains

Most nuclease superfamilies have been widely studied. Accordingly, we recapitulated the previous annotation of all distinct nuclease branches such as the phosphodiesterase superfamily, 5'->3' FLAP nuclease superfamily, 3'->5'nucleases, the bacterial endonuclease IV and V, RecB, and UvrC families. Among the many annotated DNA nucleases expressed in humans, we uncovered eight putative nucleases candidates, including: an ERCC4 (XPF) type nuclease C1ORF146 (recently designated SPO16 in mouse) (Fig. 5a, c), two DDE superfamily endonucleases GVQW3 (FLJ37770) and C21ORF140 (FAM243A), as well as Metallo β-lactamase domains in the three proteins MAP1A, MAP1B, and MAP1S (Supplementary Fig. 6a). Of the eight putative nucleases candidates retrieved from the human proteome (Fig. 1b and Supplementary Fig. 1a, b), we decided to validate the predicted C1ORF146 ERCC4 (XPF) nuclease in more detail. We chose C1ORF146 because the ERCC4 (XPF) family comprises endonucleases belonging to the flap nuclease family invariably implicated in DNA repair processes such as nucleotide excision repair (NER), DNA interstrand cross-link repair (ICL) repair, and in resolving branched DNAs structures[57]. Recently SPO16 in mice was found to bind its ERCC4 (XPF) nuclease paralog SHOC1 to function in meiotic recombination[58], raising the possibility that the SPO16-SHOC1 complex operates in a similar manner as known ERCC4 paralog heterodimers (e.g. ERCC1-XPF,FANCM-FAAP24, and MUS81-EME1). Hence, although their mutual binding interface was not conclusively mapped, it is possible that SPO16-SHOC1, and by extension, the predicted C1ORF146-SHOC1 complex in humans, might form a physical heterodimer complex analogous to related family members. To test this idea in more detail and validate whether they form a dimeric complex, we mapped the predicted C1ORF146-SHOC1 binding site using AlphaFold multimer-based complex structure[22]. Interestingly, we found that C1ORF146 and SHOC1 are predicted to reside in a high confidence heterodimeric nuclease complex (Fig. 5b) as judged by the low Predicted Aligned Error (PAE) for the interacting regions (Fig. 5f). This complex is much like the solved structure of the homologous yeast SPO16-ZIP2 complex (Fig. 5b) and other ERCC4 (XPF) nuclease dimers, suggesting that C1ORF146-SHOC1might have related

DNA repair functions. Further inspection of the contact sites suggested that C1ORF146 interacts with SHOC1 through many of the conserved residues in its two outermost C-terminal alpha-helix extensions (Fig.5c, d) like its homologous complex counterpart in yeast (Fig. 5e). Curiously, many of these signature contact sites in C1ORF146 are also conserved in other human ERCC4 paralogs (Fig. 5C) raising the question whether C1ORF146 might, in principle, have the propensity to bind other ERCC4 paralogs. We therefore modeled C1ORF146 in complex with each of the SHOC1 paralogs in humans. Despite C1ORF146 appearing to prefer SHOC1 relative to other ERCC4 nucleases (Supplementary Fig. 7e), AlphaFold Multimer predicts three additional high-confidence ERCC4 complexes i.e., between C1ORF146 and XPF, MUS81, or FAAP24 (Supplementary Fig. 7b–d). These results suggest that ERCC4 family members can, in principle, intermix e.g., as the cell's adaptive response to situations where one component is lost. The prediction that ERCC4-type heterodimers do not oligomerize or multimerize (Supplementary Fig. 7e), however, supports the idea that these heterodimers evolved with distinct functions in DNA repair. We also tested the complex formation between ERCC4-type nucleases and non-related nuclease domains (but with an overall similar 3D structure) and found that none of the unrelated nuclease families were predicted to form complexes with ERCC4-type nucleases.

In agreement with the reported function of C1ORF146 in meiotic recombination, publicly available RNA-seq data suggests that C1ORF146 mRNA is primarily expressed in testis[58] concomitant with its role in meiosis. We noticed, however, that C1ORF146 mRNA also appears to be expressed at modest levels in other organs in humans such as the brain, pancreas, skin, blood, and retina (https://www.proteinatlas.org/ENSG00000203910-C1orf146/tissue). This opens the idea that C1ORF146 might have functions other than that in meiotic recombination. Virtually all ERCC4 (XPF) family nucleases are well-described DNA repair proteins and most members function in DNA interstrand cross-link (ICL) repair[57] suggesting that C1ORF146 might share similar functions in DNA repair. To explore this possibility, we expressed human green fluorescent protein (GFP)-tagged C1ORF146 in U2OS cells to test whether C1ORF146 could recruit to sites of DNA damage after treatment with various DNA-damaging drugs. Interestingly, treatment of cells with HU, $H_2O_2$, or cisplatin led to a marked chromatin retention of GFP-C1ORF146 (Fig. 5g), indicating that C1ORF146 might recruit to damaged DNA to function in the DDRs. Supporting this idea, immunofluorescence microscopy analysis of GFP-C1ORF146 expressed in U2OS and RPE1 cells found that in a subset of cells, C1ORF146 forms discrete nuclear foci after treatment with cisplatin (Fig. 5i), suggesting that C1ORF146 has functions in DNA repair. This function might rely on active replication fork progression as C1ORF146 recruitment to chromatin is confined to S-phase cells (Fig. 5h).

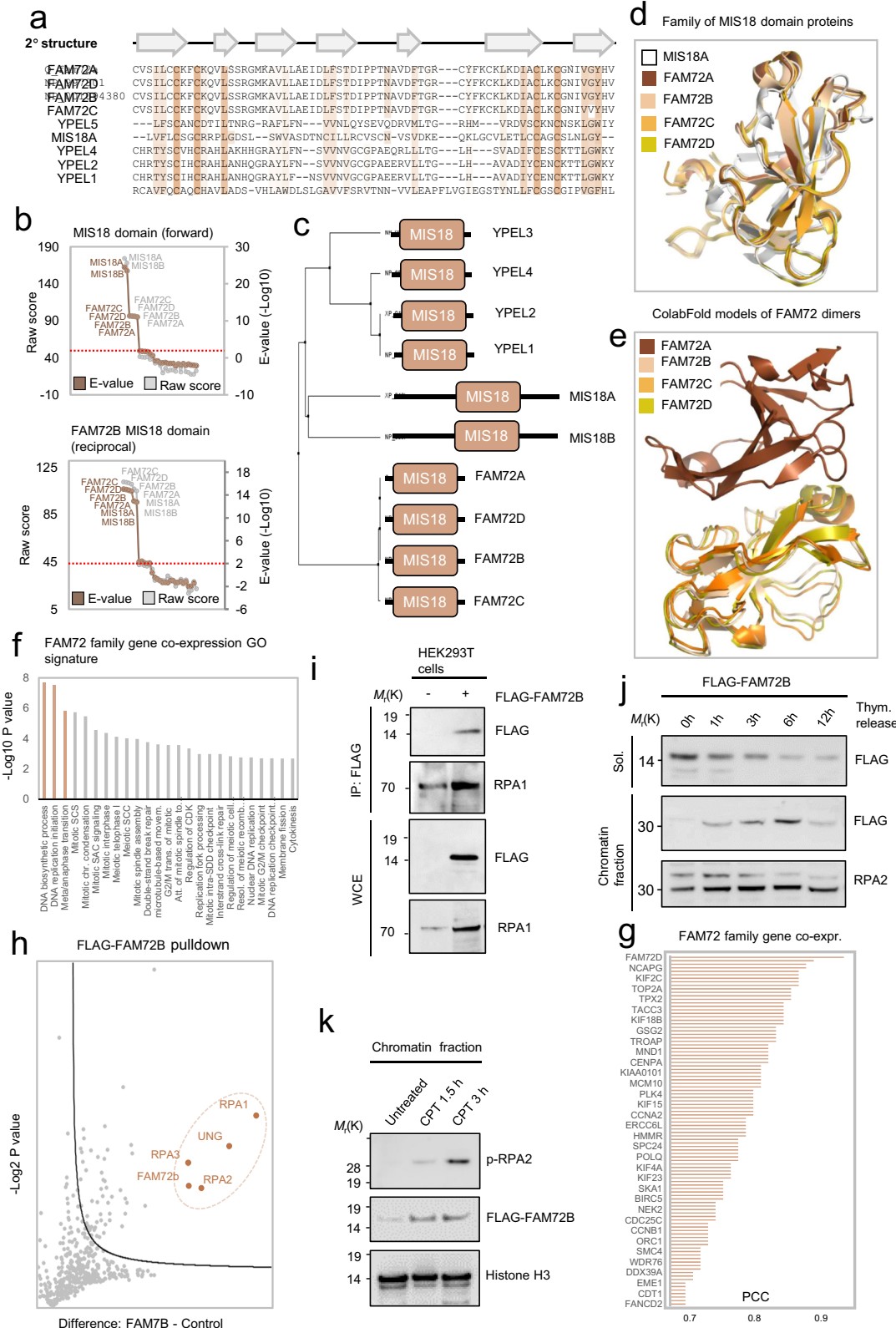

## Discussion

In this study, we utilized a combination of systematic profile-HMM-based remote homology searches and the deep learning technology of AlphaFold to explore genome maintenance protein structures across species, with the aim of discovering previously unknown domains in the human proteome. Indeed, we identified unknown evolutionary conserved modules in GM protein structures and in uncharacterized proteins indicating that many facets of human GM remain unappreciated. We identify multiple DNA binding domain candidates such as OB-fold, SAP, MYB (SANT), and KU as well as adapter domains such as TUDOR, BRCT, NNCH, POLO box, MIS18, and UBA domains, supporting the idea that such protein domains tend to exhibit low evolutionary conservation, likely to enable variability in their binding affinities. Among the proteins functioning in mitosis, we expand our previous

**Fig. 4 | FAM72 family proteins bind the RPA complex and is implicated in RPA activation in response to DNA damage. a** MSA of human MIS18 domain sequences. Conserved residues shown in brown were assessed using the Clustal W algorithm. Predicted secondary structures are shown above the MSA. Arrows indicate beta-sheets. **b** Probability plots of profile-HMM remote homology searches using either the MIS18 domain of MIS18a (forward search) or the predicted MIS18 domain of FAM72B (reciprocal search) as sequence queries. **c** Family of human MIS18 family proteins. Phylogenetic trees were calculated from MSA average distances using the percentage identity (PID) algorithm. **d** Tertiary structures of FAM72 family proteins as predicted by AlphaFold. Predicted domains were superimposed in PyMol. **e** FAM72A complexes with either FAM72B, FAM72C, or FAM72D as predicted by ColabFold[106]. **f** Gene co-expression GO enrichment analysis result of the co-expression signature profile shown in (**g**). Combined *FAM72A-D* gene co-expression signature. The human *FAM72* family co-expressed genes are

ranked according to Pearson correlation coefficients (PCC) as shown. **h** Volcano blot showing top interactors of FLAG-FAM72B as assessed by mass spectrometry. **i** FLAG-FAM72B immunoprecipitation and subsequent immunoblot of eluted immunocomplexes. Proteins were probed with the indicated antibodies. WCE = sample processing control. **j** Immunoblot of FLAG-FAM72B-expressing U2OS cells chromatin fractions after a thymidine block. Cells were either left untreated or treated for 24 hours with thymidine followed by extensive washing, release in growth medium, and harvested at the indicated time points. The resolved proteins were probed with the indicated antibodies. Sol = soluble fraction, Chromatin = chromatin enriched fraction. **k** Immunoblot of chromatin fractions from FLAG-FAM72B-expressing cells after exposure to CPT for 1.5 and 3 h. Proteins were probed with the indicated antibodies. Immunoblots are representative results of two individual experiments (X = 2). Source data are provided as a Source Data file.

discovery of the NNCH family proteins with CEP44, HAUS3, HAUS6, HAUS7, and TEDC1. Interestingly, all are associated with the network of centrosomal proteins adding further credence to the notion that the NNCH family evolved to function in mitotic and ciliary processes[26,60-63]. The divergent SAP domain is most represented among all the domains in this study, which could be due to its unique and less conserved nature compared to other protein domains. In case of the OB-fold domain our study revealed (at the time of the analysis) six unknown OB-fold-bearing human proteins. Since then, structural characterization of OB-fold domains in CXORF57, SHLD2 (FAM35A), HROB (C17ORF53), and SPATA22 have been reported[15-17,39,41,45,64-73], while SPIDR remains structurally uncharacterized. Notably, we predict SPIDR to have three OB folds in its C-terminus, resembling other ssDNA repair proteins. Indeed, our experimental evidence supports SPIDR's binding to ssDNA through its OB-fold region and point mutations affecting this domain markedly reduce ssDNA interaction. Interestingly, causal SPIDR variants implicated in primary ovarian insufficiency (POI) are predicted to disrupt OB-fold regions, affecting chromatin association. In agreement with these reports, we find that truncating mutations mimicking POI disrupts SPIDR's chromatin affinity, supporting that POI is caused by loss of SPIDR's DNA-binding properties and in turn its DNA repair activity. Curiously, HROB (C17ORF53), among other DNA repair proteins, was recently shown to be also mutated in POI[74], further supporting the notion that POI is, at least in part, a DNA repair deficiency syndrome.

Another putative predicted DNA binding module, the MIS18 (Yippee) domain family of FAM72A-D, we also found to accumulate on chromatin during DNA replication and in response to genotoxic insults, suggesting a role in DNA damage response in addition to its reported roles in recombination class switch[75,76]. Further investigation demonstrated that this function of the FAM72 family members might be aided through multimer complexes, with FAM72A-D capable of homodimerization and heterodimerization. The role of FAM72 members on chromatin might be facilitated by additional DNA repair factors as FAM72B was identified to bind avidly to UNG in addition to RPA subunits involved in DNA repair. FAM72A was previously also shown to bind[77] and suppress UNG to inhibit base excision repair during CSR in human B cells[75,76]. Hence, since RPA is known to coordinate and reside in complex with UNG during CSR[78,79] and pre-replicative repair of mutagenic uracil in ssDNA[80], FAM72B might suppress BER by acting on either UNG or RPA. Our present functional validation experiments with cells depleted of FAM72B revealed induced phosphorylation of RPA1 on serine residues 4 and 8, consistent with the functional link between FAM72B and RPA in avoiding or dealing with replication stress, a condition implicated in diverse pathologies including cancer[81].

More surprising was the identification of eight hitherto unknown domains of the PARP and PARG families considering the past year's intense scrutiny of PARPs in cancer therapy[82]. PARylation is a protein

modification that is mediated by poly(ADP-ribose) polymerases (PARPs). Many PARPs have functions in GM processes particularly PARP-1 and PARP-2, the founding members of the PARP family, have been widely studied and have well-established functions in DNA integrity surveillance. The identification of an N-terminal PARP domain in TEX15 already known to function in the DDR[31], suggests that additional PARPs partake in the chromatin remodeling after DNA damage. Recently, the PARP domain of TASOR was experimentally determined and subsequently demonstrated to be catalytically inactive. It was further found to be dispensable for assembly and chromatin targeting of TASOR but critical for its epigenetic regulation of repetitive genomic targets. Hence, TASOR was concluded to be a multifunctional pseudo-PARP that directs HUSH assembly and epigenetic regulation of repetitive genomic targets[33]. As such, the TASOR PARP domain recapitulates the non-catalytic pseudo-PARP function of ZAP (PARP13) by binding, albeit with low affinity, to RNA[83,84]. Indeed, the HUSH complex was recently demonstrated to collaborate with the NEXT complex of the RNA exosome pathway in the mechanisms of transcriptional and post-transcriptional control to limit the genotoxic activity of transposable element (TE) RNA[85]. Since we found that the predicted PARP domain in TASOR2 resembles that in TASOR, including the degenerate active site with its evolutionary loss of residues involved in NAD+ binding (otherwise conserved in active PARPs)[33], we predict that TASOR2 might exert pseudo-PARP functions similar to TASOR e.g., bind RNA species.

We also identify thus far unknown HEAT or ARM repeats and TPRs in the Integrator complex subunits. While well-defined at the functional level, the structure of the Int complex and its subunits had remained elusive until very recently. Recent cryo-electron microscopy analyzes of the human Int core complex, however, revealed that its subunits consist mostly of alpha-helical arrays adopting alpha-solenoid structures where short patches within each integrator subunit, INTS1, 3, 4, 7, and 8, could be evolutionarily annotated to known sequence repeat subfamilies[28,86]. We extend the structural features of these Integrator complex subunits to include the INTS2 and INTS10 subunits that are comprised of disparate arrays of TPRs units (Supplementary Fig. 3a, b). Thus, the Int complex represents the third network in the overall human DNA repair system, besides the ATM or ATR kinase signaling network and the Fanconi anemia pathway, to be composed of aggregates of alpha-solenoid proteins[87-90].

In conclusion, our study showcases the robust potential of computational methods in advancing our understanding of complex biological systems and demonstrates the power of combining these methods with experimental approaches for discoveries, thereby providing a roadmap for the identification of domains. Given the disease associations of mutations or deregulation of the genome caretakers[81,91,92], the approach that we illustrate in this work can also inspire the identification and exploration of potential therapeutic targets.

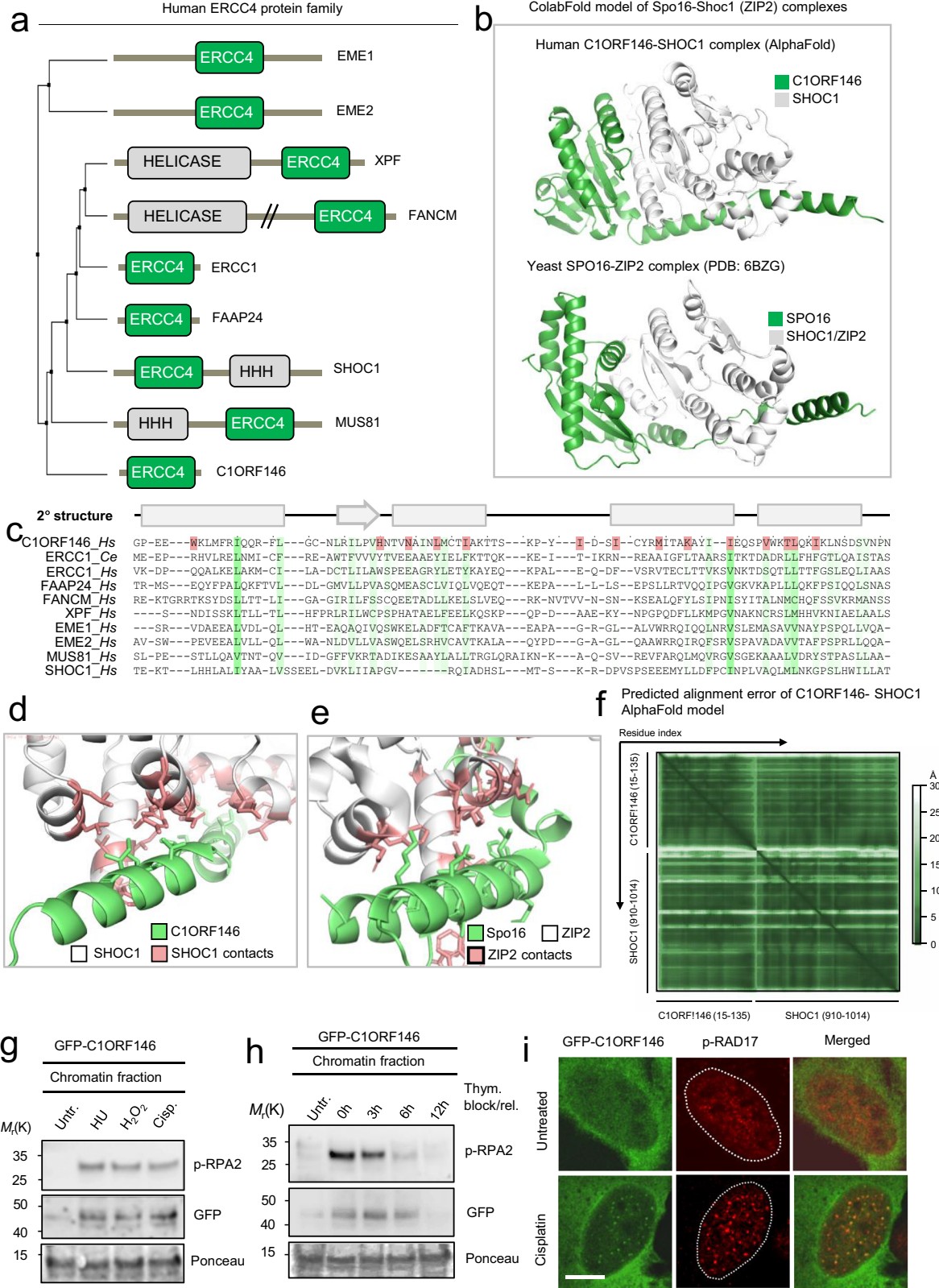

**a** Human ERCC4 protein family

**b** ColabFold model of Spo16-Shoc1 (ZIP2) complexes

Human C1ORF146-SHOC1 complex (AlphaFold)

Yeast SPO16-ZIP2 complex (PDB: 6BZG)

**c**

**d**

**e**

**f** Predicted alignment error of C1ORF146- SHOC1 AlphaFold model

**g** GFP-C1ORF146

**h** GFP-C1ORF146

**i**

## Methods
### Data collection
We compiled all annotated DDR and mitotic proteins from species for which primary evidence for protein DNA maintenance functions, through large-scale screens or low throughput studies, were available. These were extracted and filtered from commonly used databases[93] (Online Methods) and comprised four pooled gene ontology (GO) terms "DNA repair" (GO: 0006281), "DNA damage" (GO: 0006974), "cytokinesis"(GO: 0000910) and "mitotic cell cycle" (GO: 0000278) from each of the following species: *E. coli, S. cerevisiae, S. pombe, A. thaliana, C. elegans, D. melanogaster*, and *H. sapiens*. This strategy yielded a total of 3635 unique genome maintenance gene products. Next, to assess putative DDR proteins not accounted for by ontologies, for each species we examined all high-confidence

**Fig. 5 | Prediction of the C1ORF146-SHOC1 complex and implications of C1ORF146 in the DDR. a** Family of human ERCC4/XLF nucleases. Phylogenetic trees were calculated from MSA average distances using the percentage identity (PID) algorithm. **b** ColabFold[106] prediction of C1ORF146 in complex with SHOC1 and yeast SPO16-ZIP2 complex. **c** MSA of human ERCC4 domain sequences. Conserved residues shown in green were calculated using the Clustal W algorithm. Predicted secondary structures are shown above the MSA. Boxes indicate alpha-helices and arrows indicates beta-sheets. C1ORF146 residues predicted to make contacts to SHOC1 are highlighted in red. Many of these residues are conserved across ERCC4 nucleases (green) **d** Contact sites (red) between the outermost C-terminus of C1ORF146 (green) and SHOC1 (white). **e** Contacts (red) between the SPO16 outermost C-terminus (green) and ZIP2 (white). Chromatin retention of GFP-C1ORF146 in RPE1 cells. **f** The predicted alignment error (PAE) plot of the human C1ORF146-SHOC1 ERCC4 nuclease complex as shown in panel (**b**). **g** Immunoblot of chromatin fractions from GFP-C1ORF146-expressing U2OS cells either untreated or treated with DNA damaging agents. The resolved proteins were probed with the indicated antibodies. **h** Immunoblot of chromatin fractions from GFP-C1ORF146-expressing U2OS cells released from a thymidine block. Cells were either left untreated or treated for 24 hours with thymidine followed by extensive washing, release in growth medium and harvested at the indicated time points. The resolved proteins were probed with the indicated antibodies. Immunoblots are representative results of three individual experiments (X = 3). **i** Immunofluorescence microscopy images of U2OS cells expressing GFP-C1ORF146 either untreated or treated with cisplatin for 6 hours. After fixing the cells in PFA cells were permeabilized, BSA-blacked stained with the indicated antibody. The micrographs are representative of two individual experiments (X = 2). Source data are provided as a Source Data file. Scale bar in i 10 mm.

interactions in two large interaction networks i.e., physical protein-protein interactions and co-expressed genes for associations. DNA repair genes or proteins interaction data sets that comprised all physical protein-protein interactions from the Integrated Interactions Database (IID)[94], that were observed in at least two independent studies (390 GM interactors) and gene co-expression profiles (2820 GM co-expressed genes). In total, sequences from 6845 gene products, known or likely to be involved in GM processes were analyzed using the outlined profile-to-profile search scheme (Fig. 1a). Each protein sequence was used in subsequent forward or reciprocal profile-HMM searches and the compiled list of GM candidate domain structures were validated using AlphaFold2[22] (Supplementary. Fig. 1a, b).

## Computational retrieval of DDR protein interactors

In order to identify potential gene pair candidates, we explored a state-of-the-art protein-protein interaction database, along with correlated expression patterns in gene expression data. We used the Integrated Interactions Database (IID)[94]. IID is a protein-protein interaction database that integrates data from different sources and includes pairs of interacting proteins that were either identified through experiments, inferred by orthology, or predicted. As we were interested in potential candidates involved in DNA repair, we searched IID for pairs of proteins of which one was known to be involved in such a process while the other wasn't. To isolate these protein pairs of interest, we used Gene Ontology (GO) to annotate the protein pairs, focusing in particular on two GO terms: GO:0006281 ("DNA repair") and GO:0006974 ("cellular response to DNA damage stimulus"). We integrated the information available in IID by associating the GO terms for each of the two genes whose proteins take part in every interaction, taken from the GO Consortium annotation for the human genome (generated on May 22, 2018; GO version 2018-05-14). We further elaborated this annotation by adding either one of both of our GO terms of interest (GO:0006281 or GO:0006974) when at least one of their respective child terms was present. We filtered the IID database, keeping those protein pairs for which GO:006281 was present in the annotation of one of the two, but neither GO:0006281 nor GO:0006974 were present in the annotation of the other. We further filtered the resulting database, aiming at keeping only those entries that i) had at least a piece of experimental evidence and ii) had at least two associated publications containing experimental evidence of the interaction to limit the number of potential false positives. Finally, we further filtered the dataset to exclude uninteresting proteins. We considered, for each protein pair, the one of the two that wasn't annotated with GO:0006281, and filtered out the pair if the respective gene was present in a housekeeping gene list[95]. We similarly filtered the interaction list considering a list of protein contaminants frequently found in proteomics experiments, taken from the Crapome database[96].

## Analysis of gene co-expression profiles

For the initial GM gene compilation, we analyzed gene expression profiles from a panel of tissues of healthy individuals obtained from the Genotype-Tissue Expression (GTEx) database, aiming at identifying protein whose expression profiles were correlated, as proteins that are expressed with similar patterns are more likely to be interacting. We used the Co-expression Modules Identification Tool (CEMiTool)[97] to identify expression modules, i.e., groups of correlated genes, and considered each gene pair found in each of the identified modules as potentially interacting. Gene pairs were then processed by filtering them by a number of tissues they were found in, annotating them, filtered them by GOs of interest, and further filtered following the same protocol as the one we used for IID. We finally kept those pairs only present in at least two different tissues, yielding a total of 2820 pairs. Other than CEMiTool, the filtering and database manipulation was performed using Python. GOA terms annotation and ontology exploration were performed using the Biopython or goatools packages. Database processing was performed using the pandas package. All the software packages used are freely available. Scripts and data are available on our GitHub repository (https://github.com/ELELAB/DDR-candidates).

Focused gene co-expression analysis of specific genes i.e. *FAM72* family and *M1AP* genes, were performed using the Gene Expression Profiling Interactive Analysis 2021 (GEPIA2) resource based on deconvolution analysis of the normal samples from The Cancer Genome Atlas (TCGA) and GTEx databases.

## Profile-to-sequence and profile-HMM searches

For each protein, full-length FASTA sequences were extracted from the UniProt database and used to build MSAs by means of multiple re-iterative HHblits[21] or PSI-BLAST[98] searches with an E-value cutoff for MSA generation of E = 0.01. pBLAST and Iterative searches PSI-BLAST were performed at The National Center for Biotechnology Information (NCBI) http://blast.ncbi.nlm.nih.gov/Blast.cgi) and MPI bioinformatics toolkit (http://toolkit.tuebingen.mpg.de[1], respectively. BLAST and PSI-BLAST searches were performed in the non-redundant (NR) protein sequence database at National Center for Biotechnology Information (NCBI). Default settings were utilized in the searches. To avoid spurious results or statistical bias, only regions devoid of coiled-coils as judged by heptad repeat occurrence and regions masked for compositional complexity were used in the searches. Upon detection of such spurious regions, the MSA generation was performed again with query sequences devoid of coiled-coil or disordered regions. We tested various methods available either as web servers or stand-alone programs, such as Compass[99], COACH[100], Jackhmmer[3], or HHpred[4], and found HHpred and Jackhmmer to display the best performances (Fig. 1c). Hence, these methods were chosen for all subsequent HMM profile searches. HHpred searches were performed against the PFAMA database (http://pfam.sanger.ac.uk). To validate the matches, identified proteins in the first step were used as seeds in reciprocal profile-to-

profile database searches performed as above. Only matches that could be recapitulated significantly in this second reciprocal step were regarded as positive hits. Finally, to further validate the robustness of the reciprocal GM candidate profile-HMM searches, we reassessed GM candidate searches by manually inspecting MSAs for corrupted regions using the filter options for HHblits query MSAs provided by the HH-suite package. Briefly, the initial HHblits query MSA was reduced to include a diverse set of over 30 sequences spanning various species, employing the hhfilter (-diff) option. This option preserves sequences with non-homologous segments, which often exhibit the greatest dissimilarity to the query sequence. Subsequently, the filtered MSA was further refined by removing inserts and non-homologous extensions using the "remove all insert" (-r) option, resulting in a master-slave alignment optimized for identifying non-homologous regions. After visually examining the filtered and trimmed MSA, any problematic regions, such as those containing non-homologous or short sequences, were eliminated. The resulting curated MSA was then utilized to construct an HMM for subsequent profile-HMM searches. Notably, this process consistently led to the recovery of the initially identified GM candidate structure families.

The scripts for the filtering procedure of GM protein MSAs are deposited at the https://github.com/ELELAB/DDR-candidates. Finally, we made use of the recent advances in protein structure prediction by machine learning as implemented in AlphaFold2[22] and validated the compiled list of GM candidates by protein 3D modeling.

Multiple sequence alignments were built by the MAFFT program (http://myhits.isb-sib.ch/cgi-bin/mafft)[101] (and the resulting alignment edited in Jalview (http://www.jalview.org/). The consensus of the alignment was calculated and colored according to the Clustalx color scheme. Secondary structure information and structural alignment were predicted using HHpred. The AlphaFold2 software[6] was employed for homology modeling of 3D structures. Resulting 3D model coordinates were analyzed in Pymol and Discovery Studio 3.5 Visualizer. Coiled-coil propensities of target sequences were predicted as high propensity heptad repeats in Coils (http://embnet.vital-it.ch/software/COILS_form.html)[8] using both weighted and unweighted search algorithms. Identified coiled-coil regions were further validated based on the occurrence of alpha-helical content (>50% helical) using the HNN server.

## Protein complex prediction with ColabFold and AlphaFold3

To validate remote homology matches, reciprocal 3D models of candidate homologies, we utilized a modified version of AlphaFold2 on Colab notebook using the ColabFold v1.5.5 software package available at https://colab.research.google.com/github/sokrypton/ColabFold/blob/main/AlphaFold2.ipynb for protein complexes with fewer than 1400 residues[60,6,37]. The MMseqs2 search engine[46] was used to build the MSAs used for profile-HMM searches and subsequent evo blocks. No PDB templates were employed. From AlphaFold runs, AlphaPickle was used to derive the anticipated alignment score (https://zenodo.org/record/5708709#.Y3OOpHbMKUk). As shown in the figures, all structural predictions have low PAE scores for the interacting regions, indicating a high degree of certainty in the relative positions of the subunits within the complexes. All sequences used for structure prediction have at least 500 homologs in sequence databases that are currently available. Prediction of the complex between human SPIDR OB3 domain (residue 776-866) and DNA was performed in AlphaFold 3 at https://golgi.sandbox.google.com/. The DNA sequence used was 5'-GGGATTTTCAGTTTGATTGACACC-3' and the RNA control sequence used was 5'-GGGAUUUUCAGUUUGAUUGA-CACC-3'. Modeling was performed with the presence of $Zn^{2+}$.

## Phylogenetic analysis

Phylogenetic trees were obtained using phylogenetic analyzes using the maximum-likelihood method in the FastTree 2.1 program and IQ-Tree v.2.050 to estimate phylogenetic trees. In our analyzes, we utilized molecular evolution models determined through ModelFinder[51] integrated into IQ-Tree. This approach identified an optimal partitioning scheme and the best model for each partition. To evaluate the support for the resulting topology, we conducted 1000 ultrafast bootstrap replicates[52]. The maximum-likelihood method from both the FastTree 2.1 and IQ-Tree v.2.050 programs produced essentially the same trees.

## Antibodies

For immunoblot analysis, the following primary antibodies were used (dilutions in parenthesis): Mouse anti-FLAG (F-1804, 1:500) from Sigma, rabbit anti-FLAG (F7425, 1:1000) from Sigma, rabbit anti-GFP (sc-8334, 1:500) from Santa Cruz, mouse anti-GFP (sc-9996, 1:500) from Santa Cruz, rabbit anti-RPA1 (ab79398, 1:500) from Abcam, rabbit anti-Phospho-RPA32 (Ser4, Ser8) (A300-245A, 1:500) from Bethyl Laboratories, mouse anti-RPA32 (ab2175, 1:500) from Abcam, rabbit anti-GAPDH (2118, 1:2,000) from Cell Signal, and rabbit anti-histone H3 (ab1791, 1:4000) from Abcam.

## PCR, cloning procedures, and plasmids

Plasmids encoding full-length and truncated versions of FLAG-SPIDR or full-length GFP-SPIDR were generated by PCR with relevant primers and human SPIDR plasmid as a template (Origene) followed by cloning into pFLAG-CMV2 (Sigma) or EGFP-C1 (Clontech) by standard procedures. Plasmid encoding human FLAG-tagged C1ORF146 was purchased from Origene Technologies (RC215196). Plasmid encoding full-length GFP-C1ORF146 was generated by PCR with relevant primers and human plasmid as a template (RC215196) followed by cloning into EGFP-C1 (Clontech). Mutagenesis of GFP-SPIDR by C817A and C820A substitutions was performed by a two-step PCR protocol as described previously with relevant primers[102]. Briefly, the point mutations were introduced by first performing two parallel PCR reactions using mutant primers and the same flanking primers as used to generate the WT GFP-SPIDR. First PCRs was performed using the primers

Forward: 5'-AAAAAGAATTCAATGCCCCGCGGCAGCCGC-3' (EcoRI) with reverse: 5'-AAAAAGTGACCACCCGGGAGGCGTCCCCAGCGGAAAAGGCGCCTC-3' and forward 5'-AAAAAGAGGCGCCTTTTCCGCTGGGGACGCCTCCCGGGTGGTCAC-3' with reverse 5'-AAAAAGTCGACC-TAGTGTTCTGCAGAGGC-3' (SalI). Finally, the second step was performed by PCR of the flanking forward and reverse primers mixed with the PCR products of the first PCR step. The resulting PCR fragment was digested with the respective restriction enzymes and cloned into the EGFP-C1 vector (Clonetech).

## Cell culture and transfections

HEK293T and U2OS cells were grown at 37 C in Dulbecco's modified Eagle's medium (DMEM, Gibco) with 10% heat inactivated fetal bovine serum (FBS, Gibco) and penicillin-streptomycin (Gibco) using 5% $CO_2$ and 95% humidity. The RPE1 cells (laboratory stock, derived from the immortalized hTERT RPE1 cell line, ATCC CRL-4000) were grown in 45% DMEM and 45% F-12 (Ham; Sigma) with 10% FBS and penicillin-streptomycin; cultures were passaged every 3-4 days. For plasmid transfection of HEK293T cells, 8 mg DNA was transfected into cells in a 15 cm dish using Fugene 6 and incubated for 24 hours.

For FAM72B protein depletion, U2OS cells were transfected with siRNA two times over two days and incubated additionally two days. Cells were grown to approximately 80% confluence in 9.6 cm2 petri dishes before first transfection; 5-6 hours after transfection the medium was changed. Two FAM72B-specific siRNAs were used, both purchased at Eurofins MWG: Operon: siFAM72B-1 (5'- CCA GGC AGU UUA UGA UAU U-3') and siFAM72B-2 (5'-CAG CAU GAU GUU AGA UAA A-3'). All transfections with siRNA (final concentration 250 nM) were carried out using DharmaFECT Duo. siCONTROL (Dharmacon) was used as a control siRNA 6 hours after transfection fresh growth medium was added. The cells were subjected to double transfection with an interval

of 24 hours. In Biotin-DNA pull-down assays, Dynabeads T1 from Life Technologies underwent two washes in PBS buffer before being bound to biotinylated DNA substrates at room temperature for 30 minutes. Subsequently, the beads underwent two additional washes in PBS buffer, followed by two washes in binding buffer (composed of 80 mM Tris, pH 7.5, 100 mM KCl, 5 mM MgCl2, 2 mM DTT, and 100 mg/ml BSA in RNase and DNase-free water). Next, 1ul of beads carrying 4 picomoles of bound DNA was resuspended in the binding buffer. Approximately 500 femtomoles of purified protein were introduced to the mixture, which was then rotated at room temperature for 30 minutes. The supernatant was discarded, and the beads were boiled in 2X sample buffer for 5 minutes. Subsequently, captures were subjected to analysis through immunoblotting. The single-stranded DNA oligo employed for pull-down assays was a poly dT50, while the double-stranded DNA was created through annealing 5′-Biotin-GGAT-GATGAC TCTTCTGGTCCGGATGGTAGTTAAGTGTTGAG-3′ with its complimentary oligo.

## Immunofluorescence microscopy

IFM analysis of GFP-C1ORF146-expressing RPE1 cells was carried out as follows. Cells grown on glass coverslips were washed once in ice-cold PBS, fixed with 4% paraformaldehyde (PFA) solution, permeabilized with permeabilization buffer (PBS with 0.1% (v/v) Triton-X100 and 1% (w/v) bovine serum albumin (BSA) and subjected to IFM as described previously[103]. Imaging was done using a Zeiss Observer Z1 microscope. Images were processed for publication using Adobe Photoshop CS4 version 11.0.

## Immunoprecipitation and immunoblotting

HEK293T cells were transfected the day before immunoprecipitation (IP). Cells were harvested in ice-cold EBC buffer (140 mM NaCl, 10 mM Tris-HCl, 0.5% NP-40 and protease inhibitor cocktail (Roche)). For FLAG IP experiments, cleared cell extracts were incubated 1 h with 20 µl anti-FLAG (M2) conjugated magnetic beads (Sigma) under constant rotation (4 C). The subsequent IP was performed with 10 µl Anti-FLAG (M2) conjugated magnetic beads for 1 h, and Immunocomplexes were washed five times in EBC buffer before elution with 1x FLAG peptide (Sigma). Eluted FLAG-proteins complexes were purified further by micropore filter centrifugation.

Analysis by SDS-PAGE and immunoblotting with relevant antibodies was performed using the Novex system from Invitrogen and by following the protocol supplied by the vendor. Blots were incubated in primary antibodies at appropriate dilutions, incubated with relevant horse radish peroxidase-conjugated secondary antibodies. Images were processed in Adobe Photoshop CS6.

For immunoblot analysis, the following primary antibodies were used (dilutions in parenthesis): rabbit anti- RPA2 pSer33 (A300-246A, Bethyl (1:500)), rabbit anti-RPA1 (Ab79398, Abcam) (1:500), rabbit anti-FLAG (1:1000) from Invitrogen, mouse anti-FLAG-tag (F1804 (Clone M2), Sigma Aldrich (1:500)), mouse anti-GFP (11814460001, Roche (1:500), rabbit anti-GFP sc-9996 (Clone B2), Santa Cruz (1:1,500)), anti-Histone H3 (#9715, Cell Signal (1:2000). Secondary antibodies used for immunoblotting: horseradish peroxidase-conjugated goat anti-mouse (P0447, 1:4,000) or swine anti-rabbit (P0399, 1:4,000) from Dako. For IFM analysis, the following primary antibodies were used: Phospho-RAD17 (Ser656) from ThermoFisher (711717). Alexa Fluor 350-conjugated donkey anti-mouse (A-10035) or donkey anti-rabbit (A-10039); Alexa Fluor 488-conjugated donkey anti-mouse (A-21202), donkey anti-rabbit (A-21206) or donkey anti-goat (A-11055), Alexa Fluor 568-conjugated donkey anti-mouse (A-10037), donkey anti-rabbit (A-10042) or donkey anti-goat (A-11057).

## Mass spectrometry

To understand the potential function of the FAM72 paralogs in GM, we searched for FAM72B-interacting proteins as an example using the FLAG affinity purification method on cell extracts of cells expressing the FLAG-FAM72B (Supplemental Fig. 5a). The FLAG-purified immunocomplexes were eluted with FLAG peptides and prepared with the Protein Aggregation Capture (PAC) method[104] followed by Trypsin and Lys-C-digestion. Peptides were subsequently analyzed by mass spectrometry. Mass spectrometry raw data were subsequently processed in MaxQuant and was further analyzed in Perseus. The experiment was done in two replicas (X = 2) using a control and a FLAG-FAM72B pull-down per experiment (four samples in total). The specific details of the mass spectrometry analysis were as follows. The nLC-nESI MS/MS analysis was performed in an Easy1200 chromatographic system coupled to a Exploris 480 mass spectrometer (Thermo). About 1 µg of the tryptic digest was applied into an analytical column (New Objective x 75 µm internal diameter x 1.9 µm particle size ReproSil-Pur 120 C18-AQ, DR. MAISCH). Mobile phase A (0.1% v/v formic acid in water) and mobile phase B (0.1% v/v formic acid in acetonitrile) were used in a separation gradient from 2 to 40 % B for 120 minutes. The spray voltage was adjusted to static in the nanoelectrospray source, with no auxiliary gas flow and capillary temperature are also static. The lens voltage was set to 50 V. MS1 spectra were acquired in the profile mode in the Orbitrap analyzer (m/z 350 to 1500) with a resolution of 60,000 FWHM and Automatic Gain Control % (AGC) set to 300. Up to 12 precursor ions per MS1 spectrum were selected for fragmentation with higher-energy collisional dissociation (HCD) with normalized collision energy (NCE) of 30. The isolation window was set to 1.2 m/z and the dynamic exclusion configured to 60 s. MS2 spectra were acquired in the Orbitrap at a resolution of 30,000 FWHM; AGC (%) was set to 200, intensity threshold of 10,000 counts. Singly charged and unassigned ions were not subjected to fragmentation. Data were obtained using Xcalibur software (version 4.4.16.14).

## Bioinformatic analysis of proteomic data

The mass spectrometry data was searched using MaxQuant version 2.0.3.0 which includes the search andromeda integrated with default parameters, against human FASTA file downloaded from UniProt (uniprotkb_human_proteome_AND_reviewed_t_2023_06_18). Briefly, the parameters used for the search included carbamidomethylation of cysteines as a fixed modification and the variable modifications, methionine oxidation, and N-terminal acetylation. The criteria for peptide acceptance were peptide PSM FDR 0.01, protein FDR 0.01, 10 ppm for peptide tolerance, peptide minimum length 7 amino acids, peptide maximum length 45, site decoy fraction 0.01, minimum peptide 1, minimum razor + unique peptides 1, minimum unique peptide 0, minimum score for unmodified 0, minimum delta score for modified peptides 6, main search maximum combinations. Digestion mode was specific for trypsin with maximum 2 missed cleavages. The MaxQuant output data was further analyzed in Perseus version 2.0.10.0 using the LFQ intensities filtered for contaminants, reversed peptides, and proteins only identified by site. Log2 transformed LFQ intensities with imputed missing values were subjected to two-sample t-test followed by volcano plot visualization.

## Reporting summary

Further information on research design is available in the Nature Portfolio Reporting Summary linked to this article.

## Data availability

Proteomics data have been deposited to the ProteomeXchange Consortium via the PRIDE partner repository with PXD identifier: PXD043273. The following structures from the PDB database have been used: 6BZG (yeast SPO16-ZIP2 complex), 1JEY (human Ku heterodimer bound to DNA), 3FA2 (human BARD1 Tandem BRCT Domains), 7KK2 (human PARP1 catalytic domain), 6UEJ (human PARP13 bound to RNA), 4B1G (human PARG catalytic domain), 6I52 (yeast RPA bound to ssDNA), 7SFZ (human Mis18a-yippee domain),

6G70 (murine Prpf39), 2LLK (human DMTF1 MYB (SANT) domain), and 1JJR (SAP domain of human KU70). Validation reports for each structure are provided for the respective PDB entries at https://www.rcsb.org/. Datasets for gene co-expression analysis are available http://gepia2.cancer-pku.cn/#dataset and the Cancer Genome Atlas (TCGA) https://www.cancer.gov/ccg/research/genome-sequencing/tcga. For the initial data compilation and filtering, the house keeping list was retrieved from https://doi.org/10.1016/j.tig.2013.05.010 [95]. Bulk raw output data can be found in the Figshare repository (https://doi.org/10.6084/m9.figshare.26014669.v1, https://doi.org/10.6084/m9.figshare.26014621.v1, https://doi.org/10.6084/m9.figshare.26015185.v1, https://doi.org/10.6084/m9.figshare.26014609.v1, https://doi.org/10.6084/m9.figshare.26014858.v1, https://doi.org/10.6084/m9.figshare.26014900.v1, https://doi.org/10.6084/m9.figshare.26015095.v1, https://doi.org/10.6084/m9.figshare.26015680.v1, https://doi.org/10.6084/m9.figshare.26015332.v1). Source data are provided with this paper.

## Code availability
The scripts, input files and output files are available at GitHub: https://github.com/ELELAB/DDR-candidates.

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

## Acknowledgements

This work was funded by the following grants from: the Danish Cancer Society (R322-A17482), the Swedish Cancer Fonden (nr. 170176), the Swedish Research Council (VR-MH 201446602-117891-30), the Novo Nordisk Foundation (NNF 20OC0060590), Danish Foundation for Independent Research (DFF 1026-00241B), the Danish National Research Foundation (project CARD, DNRF 125) and Carlsberg Foundation Distinguished Fellowship (CF18-0314).

## Author contributions

K.B.S. and J.B. designed and interpreted experiments. K.B.S., S.M., M.Tahir., and A-S.N. conducted experiments. K.B.S., M.Tiberti. N.T., E.P. designed and carried out bioinformatics analysis. K.B.S. wrote, and J.B. and J.S.A. modified the manuscript. J.B. directed the project. All authors commented on and approved the manuscript for submission.

## Funding

## Competing interests

The authors declare no competing interests.
