## [Peer Review File · Nature Communications]

Exploring the structural landscape of DNA maintenance proteinsREVIEWER COMMENTS

Reviewer #1 (Remarks to the Author):

The work presented in the manuscript identifies novel genome maintenance (GM) proteins, based on computational predictions followed by a selected candidate experimental validation. Overall, is a very descriptive and comprehensive catalogue of the domain repertoire of GM proteins. They present novel findings in terms of extending the already known protein domain repertoire of GM proteins, and an experimental validation of certain candidates.

I have two main concerns regarding the original assumptions of the paper:

1) Protein Function does not always follow the orthologue's conjecture as they could perform very differently (see for instance 10.1038/srep30443), and 2) that we can infer evolutionary relationships at that level of sequence (dis)similarity, based on structural conservation. We cannot. There is plenty of evidence that at the twilight zone, even if we can grasp relationships between profiles (remote homology), no evolutionary relationships can be assigned based on fold similarity. At this level of sequence similarity, the signal is lost, and convergent evolution events could arise.

This is precisely the reason why remote homology-based methods are not used regularly to evolutionary annotate proteins. Summarizing, at this level of divergence, we cannot assess for certain evolutionary relationships, nor function without a proper experimental validation.

I also have methodological concerns:

1. The starting dataset is based on an starting retrieval of proteins annotated with certain GO terms, followed by screening of Protein Protein Interaction repo, and further curated with co-expression networks to enrich the dataset. Some phylogenetic analyses are conducted.

How redundant/diverse is this final dataset? How many species are represented there? What are the associated GO terms which are not related to your interest?

2- Regarding the GO terms analysed in the work. Terms were selected according to which category? The GO: 00006281 is nowhere to be found in GO-related databases. For instance, a quick search gives five associated results <https://www.ebi.ac.uk/QuickGO/searchterms/GO:00006281>.

Ideally, the results presented in the submitted work, should stand in the light of the current GO graph. Perhaps an alternative way is to select also proteins is to extract the GO terms from the protein domains of the list, and extract those more specific for each category.

3- Regarding the PPI, there is a recently release dataset of PPI devoted to DDR, which I believe to be more adequate to use for the screening purposes of this paper, <https://doi.org/10.1016/j.cels.2023.04.007> . Ideally, these should constitute a more curated dataset to be used for screening, as it provides assemblies.

4- The co-expression networks are by default extremely noisy (10.1186/s13059-019-1700-9). There are many reasons why genes can be co-expressed to together, and not being involved in a particular function. It is not clear how confounding effects have been removed from the "standard pipeline" from TCGA biolinks. There are more adequate methods, alternative to Spearman correlation, see for instance: 10.1093/bioinformatics/btab041.

5-The pipeline, the authors put a lot of effort to run many methods to achieve the same thing. For instance, they initially use HHblits and PSI_BLAST for remote homology. Since it is long known that PSI-BLAST provides a great deal of false positives (more than HHblits), I failed to grasp the relevance of running PSI-BLAST, unless the provide the relative contribution of each one for the identification of the domains, which is not given in the paper.

6- They use Alpha Fold to predict the structures, in its multimer conformation. Some details are missing, for example, are the previously generated MSA fed into the pipeline (AF2 generates HHBLITS itself to run the evoblocks)? Are pdb templates also used? Those are relevant issues. Why is not Foldseek used systematically in the identified proteins?

Line 237. All paralogues tend to keep the same structure, I do not grasp the relevance of the sentence. They could participate in the same pathway, doing different things.
line 422, where does the house keeping gene list come from?
line 429, the github repository to evaluate the code is not accessible, so reliability and reproducibility of the code cannot be evaluated.
line 453, IQtree is a better choice of making phylogenies from distantly related sequences, and provides real bootstrap values.

line 544, typo two dots.
line 553, A clarification: fold similarity does not mean necessarily to be evolutionary related nor to be functionally similar.
line 579, the sentence about Clustal is confusing, it reads as the MSA is generated using this method.

Reviewer #2 (Remarks to the Author):

Exploring the structural landscape of DNA maintenance proteins
KB Schou et al.

The paper describes a systematic survey of DNA maintenance proteins from bacteria to humans using profile-to-profile models. The study aims to identify new DNA maintenance protein candidates and annotate new domains in established DNA maintenance proteins.

The researchers collected annotated DNA damage response (DDR) and mitotic proteins from various species and used profile-to-profile search schemes to analyze the protein sequences. They applied iterative profile-HMM searches and validated the candidate domain structures using AlphaFold. Through their analysis, they identified several new conserved domains and expanded the understanding of known protein families.

Some of the key findings of the study include:

- Discovery of new DNA binding domain families: The analysis uncovered previously unknown members of DNA binding domain families, such as MYB/SANT, TUDOR, OB-fold, Mis18/yippee, SAP, WSD, and KU70/KU86 beta-barrel domains.
- Identification of new mitotic protein family members: The study identified new members of six protein families involved in mitotic functions, particularly kinetochore-related proteins.
- Annotation of new conserved domains in DNA repair proteins: The analysis revealed new conserved domains in DNA repair proteins, including the Integrator (Int) complex subunits and potential poly(ADP-ribose)-related enzymatic processes.
- Expansion of the understanding of DNA-binding domains: The study expanded the superfamily of MYB/SANT domain-containing proteins and predicted new SAP domains in various proteins, including PARP1.

Overall, the study highlights the use of computational methods to explore the structural landscape of DNA maintenance proteins and discover new domains and relationships. The findings provide valuable insights into the repertoire of genome stability caretakers and contribute to a better understanding of DNA maintenance processes.

The following comments are suggested to improve the paper.

1. The technique of Profile HMMs has existed for a long time. The authors should cite literature that supports their notion that GM proteins have not been subjected to systematic comparative analysis by computational strategies and some of the reasons as to why.

2. The work needs to mention prior progresses made in this field (GM proteins) using computational means (to provide a bit of background) and where this work picks up from and the approach used here to produce these findings (a high-level discussion).

3. The excerpt lacks a clear and well-structured introduction. It would be beneficial to provide a more concise and organized overview of the research question, methods employed, and key findings. This will enhance the readability and comprehension of the paper for the intended audience.

4. Some recent techniques like DeepInsight (PMID: 31388036) and Transformer model (<https://arxiv.org/abs/1706.03762>) are used in extracting relevant features for deep learning and/or machine learning methods. It would be good to describe these techniques (DeepInsight and Transformer model) and explain why the profile-profile HMM model was selected.

5. The excerpt mentions the use of profile-to-profile models and iterative profile-HMM searches, but it lacks specific details on these methods. Providing a more detailed description of the computational strategies employed would improve the reproducibility and clarity of the study. In particular, in lines 438 to 446, a reader would like to know what are the results/statistical bias in more detail, more description of coiled-coils and heptad repeat analysis. Providing details will be beneficial. Also in line 441 and 442, the best performance using those methods are not presented.

6. Validation and Experimental Results: The excerpt briefly mentions experimental validation of the predicted domains in SPIDR and FAM72 protein families. However, it does not provide specific details about the experimental methods used, the results obtained, or any statistical analyses performed. Including these details would strengthen the paper's scientific rigor and credibility.

7. Discussion and Interpretation: The excerpt briefly mentions the identification of new domain families and their potential implications but does not delve into a comprehensive discussion or interpretation of the findings. The authors should provide a more in-depth analysis of the functional implications of the newly identified domains and their potential roles in DNA maintenance and repair processes.

8. Language and Writing Style: The excerpt contains several grammatical and typographical errors that need to be addressed. Additionally, the writing style can be improved to enhance clarity and coherence. Proofreading the manuscript thoroughly and revising the language to ensure a clear and concise expression of ideas would be beneficial. Some examples are:

a). Page 2, Line 40: typo "fully5"

b). Page 2, Line 61: Grammar "... identified in an multiple sequence ...". 'an' should be 'a'.

Reviewer #3 (Remarks to the Author):

In this manuscript, Schou et al. report the identification of a large number of new human proteins involved in the DNA damage response (DDR) and genome integrity maintenance. The study is based

on a sophisticated bioinformatics strategy, complemented by robust proteomics and some cellular biology and biochemistry of selected examples of the presented proteins, to validate this approach. The manuscript provides a significant advance and would likely be of interest to a broad audience of researchers in several fields. In my opinion, the most valuable parts of this study are twofold: a) The well explained innovative bioinformatics strategy that in itself should be also applicable to other fields of biomedicine and hence help discover many more proteins (related to other cellular functions) based on the principles applied here, and then b) The 'catalogue' of the new proteins (and domains) itself, the future detailed functional characterization of which will most likely be taken up by many research labs with interests in the individual protein families whose new members are presented in this manuscript. The work is within the scope of Nature comms, it is overall a good candidate for the journal, and its character makes it probable that it will in time be cited by other groups, thanks to its 'discovery' of new protein members of families implicated in important biological processes and pathologies. Nevertheless, there are several issues that in my opinion need strengthening to better support the conclusions and enhance the impact of this dataset before it can be fully recommended for publication.

Specific criticisms and suggestions

1. As to the powerful bioinformatics strategy presented here, it could still be improved to some extent: Thus, while the HMM-based predictions of protein domains are accurate and paved the way for the revolution in artificial intelligence-based structural predictions of protein domains by e.g., AlphaFold, it is generally recommended to preform additional statistical assessment of the identified candidates. This additional support should help to rule out the possibility of false positive candidates, and it is especially critical for validation of the shorter homology regions.
2. It would be informative to include (perhaps in the Supplementary material and for at least some of the proteins ort domains) multiple sequence alignments of candidate families to see in detail the conserved properties of the identified domains.
3. All presented DDR candidates are of human origin. It would be informative for the broader research community to include new DDR protein candidates from other species e.g., from the model organisms used to collect the initial protein domain list (at least the most some attractive protein families covered by this work).
4. I do realize that the identified protein domains and candidate proteins are too numerous to require functional validation for all, or a really deep characterization. Nevertheless, some additional functional insights into some attractive protein, such as the FAM72B presented in the manuscript, would further strengthen the conclusions and relevance for biology. For example, what phosphor-RPA species is affected by FAM72B-depletion and in turn what downstream mediators/effectors within the DNA damage network are affected? Furthermore, are the FAM72B-depleted cells hypersensitive to DNA damage?
5. Another attractive finding of the manuscript is the predicted association between C1ORF146 and other ERCC4 domain proteins, but this part would benefit from further support by experimental validation, such as through selected example(s) of co-immunoprecipitation from suitable human cell model(s). Such additional experimental data would further support the interaction predicted by the AlphaFold Multimer analysis shown in the manuscript.

Rebuttal with replies to reviewers' comments

Before we address the individual comments from each of the three reviewers, we wish to express our thanks for their valuable remarks and criticisms that have guided our efforts over the last 7 months to revise the manuscript accordingly. As a result, in addition to the suggested textual changes, a total of 20 figure panels in this revised manuscript are either newly generated or represent modified versions of previous figures (as indicated in the specific answers below) based on the new data gained during the revision process, to follow the suggestions of the reviewers. We have addressed all points raised to the best of our abilities, and we believe the resulting revised manuscript is now more conclusive, informative and readable, thanks to the thoughtful feedback from the reviewers.

In the point-by-point replies below, our responses are in italics, to be easily distinguished from the comments of the referees.

Reviewer #1 (Remarks to the Author):

The work presented in the manuscript identifies novel genome maintenance (GM) proteins, based on computational predictions followed by a selected candidate experimental validation. Overall, is a very descriptive and comprehensive catalogue of the domain repertoire of GM proteins. They present novel findings in terms of extending the already known protein domain repertoire of GM proteins, and an experimental validation of certain candidates.

I have two main concerns regarding the original assumptions of the paper:

1) Protein Function does not always follow the orthologue's conjecture as they could perform very differently (see for instance 10.1038/srep30443), and 2) that we can infer evolutionary relationships at that level of sequence (dis)similarity, based on structural conservation. We cannot. There is plenty of evidence that at the twilight zone, even if we can grasp relationships between profiles (remote homology), no evolutionary relationships can be assigned based on fold similarity. At this level of sequence similarity, the signal is lost, and convergent evolution events could arise. This is precisely the reason why remote homology-based methods are not used regularly to evolutionary annotate proteins. Summarizing, at this level of divergence, we cannot assess for certain evolutionary relationships, nor function without a proper experimental validation.

Our response:

We agree with the Reviewer that one must be cautious about the conclusions drawn from profile-HMM type searches. We also agree that the results yielded by remote homology-based methods are in the twilight zone and such relationships should be assessed experimentally. However, we believe that this reviewer is unnecessarily skeptical in his/her comment that we cannot assess for evolutionary or functional relationships using our computational approach for systematically assessing genome maintenance homologous protein domains.

First, we never stated that our search results are exact. Indeed, our aim with this in silico survey was to identify potential GM candidates in humans for the research community to study in more detail in the future. Whether our GM candidates represent evolutionarily conserved or diverted functions remains to be validated experimentally. For this reason, we

have experimentally validated a collection of candidates identified in our survey, which confirmed the predicted functions, thereby indicating a reasonable, indeed in the cases studied a robust, degree of reliability among our computational predictions.

Second, that said, we believe that the remote homologies established in our survey are in fact reliable in the vast majority of cases (if not all) and they represent genuine evolutionarily conserved families, as we will discuss below.

With regard to the doubts raised by the reviewer, we see two overall ways by which remote homology searches could result in detection of false positive structural/functional relationships:

1. By detecting distant homologies that diverted so early in evolution that the biochemical properties of the domain/module have changed and the functional relationship is lost, as the reviewer states. We can say already now, however, that while remote homologous proteins COULD have evolved different functions, often (perhaps more often), the homologous proteins (identified by profile HMM searches) indeed share similar functions. Much evidence supports this notion. Below is a list of the protein candidates initially identified in our survey (which we initiated in 2017) that have since been published by others to contain the predicted domains and function in GM, thereby validating entirely independently our functional predictions:

- The OB fold domains of **CXORF57** (DOI: 10.15252/embr.201744877, DOI:10.1016/j.molcel.2017.06.023), **FAM35A** (10.1016/j.molcel.2017.06.023, DOI:10.1016/j.molcel.2017.06.023), **C17ORF53** (doi: 10.1101/gad.329508.119), **SPATA22** (doi: 10.1093/humrep/deab185), and **TDRD3** (doi: 10.1093/hmg/ddn219)*
- The KU80 domain of **INST14** (DOI: 10.1038/s41467-020-17232-2)*
- The ERCC4/XPF domain of **C1ORF146** (DOI: 10.1126/sciadv.aau9780)*
- The PARP domain of **TASOR** (doi: 10.1038/s41467-020-18761-6)*
- The BRCT domain of **SMARCC1** (doi: 10.1038/s42003-021-02050-z)*
- The UBA domain in **N4BP1** (PMID: doi: 10.1038/s41467-021-21711-5)*
- The triple TUDOR domain of **SETDB1** (doi: 10.1038/s41467-017-02259-9.)*
- The Alba_2 domain in **Schlafen** (PMID: doi: 10.1016/j.jmfm.2019.04.003.).*
- The HEAT/ARM repeats of the integrator complex subunits: **INTS7, INTS5, INTS4 INTS2, and INTS1** (DOI: 10.1126/science.abb5872)*
- The TPR of **INTS8** (doi: 10.1126/science.abb5872.).*

Thus far all the experimentally determined structures of the candidates identified in our study have been demonstrated to agree with our predictions, thus providing a strong and independent experimental support of our predictions and conclusions.

2. Then comes the question of whether the remote homologies represent true evolutionary relationships or whether they could arise by chance from dissimilar lineages through convergent evolution. Our candidates are not identified by fold similarity. Maybe here what the Reviewer suggests is that because remote homology searches have been published to falsely detect relationships among proteins, the profile-HMM method might be generalized to perform little better than common protein fold similarity algorithms like e.g., the DALI fold prediction software. If the latter was the case, we would agree with the Reviewers concern that “no evolutionary

relationships can be assigned based on fold similarity” since fold similarity comparison does not provide information about evolutionary or functional relationships across proteins. In addition to the published experimental evidence outlined above, we have additional reasons to believe, however, that our profile-HMM searches detect genuine evolutionary relationships, as explained below. To rule out any likelihood of spurious matches, we have now (in the revised manuscript) introduced a relatively simple yet powerful solution to the problem of convergent evolution (or homologous overextension of alignments and other pitfalls), which we have implemented in the method pipeline as illustrated in Fig. 1a. Our solution has been to conduct both forward and reciprocal profile-HMM searches using template and subsequently candidate matches as queries, and then plot the resulting match scores to visualize the distinction between genuine protein families and unrelated matches that arose due to redundant dissimilar. For most GM candidates (except for the redundant HEAT/ARM/TPR repeats), our forward (template query -> candidate match) and reciprocal (candidate match -> template query) searches confirm the initial homologies that we outlined in the first submission. Throughout the revised manuscript we have visualized the score distribution of forward (template query -> candidate match) and reciprocal (candidate match -> template query) search results as probability plots. These efficiently show the accuracy/reliability of the identified protein families compared to the bulk of insignificant matches (Fig. 1f, 2f, 3b, 4b, Supplementary Fig. S2d, S3c, S3d, S3d, S4c, S6c, and S7a). It is evident from these plots that all the candidate protein families analyzed (Supplementary Fig. S1) stand out markedly more significantly than the bulk spurious matches. If the profile-HMM searches had detected significant matches to evolutionarily unrelated sequences (but with similar fold), this promiscuity would likely manifest in multiple matches due to the redundancy of proteins with similar fold, as is often the case for fold recognition searches such as DALI. Our forward and reciprocal searches, however, consistently produce only a small number of matches suggesting that the candidates identified are genuine family members, as evident from the probability plots. In addition, it is highly unlikely that reciprocal profile-HMM searches would produce matches to the same small family of proteins as the forward searches.

- 3. In addition to the plotted E-values of the forward and reciprocal searches in the score distribution plots, we have also displayed their calculated Viterbi raw scores to provide a more direct measure of the alignment qualities of the HMM-to-HMM matches. The E-value statistic estimates the likelihood of obtaining a similar or better score by chance but contrary to the Viterbi raw score, the E-value is sensitive to the size of the sequence database being searched. In very large databases such as the PFAM database, even random and short matches might have low, yet significant E-values in cases where the raw score is also low and may suggest otherwise. The raw scores provided in the probability plots shown in Fig. 1f, 2f, 3b, 4b, Supplementary Fig. S2d, S3c, S3d, S4c, S6c, and S7a however, indicate that query-match alignments are reliable.*

I also have methodological concerns:

1. The starting dataset is based on a starting retrieval of proteins annotated with certain GO terms, followed by screening of protein Interaction repo, and further curated with co-

expression networks to enrich the dataset. Some phylogenetic analyses are conducted. How redundant/diverse is this final dataset? How many species are represented there?

Our response:

We appreciate the Reviewer's comment and realize that we should have stated our data retrieval strategy more clearly.

Our initial GM gene/protein compilation list involve three independent steps as follows:

- 1. Gathering GM GO terms from seven organisms (E. coli, S. cerevisiae, S. pombe, A. thaliana, C. elegans, D. melanogaster, H. sapiens). GO terms used: "DNA repair" GO: 0006281, "DNA damage" GO: 0006974, "cytokinesis" GO: 000910, and "mitosis" GO:0000278).*
- 2. Compiling GM protein interactors from humans only (retrieved from public proteomics data). GO terms used: "DNA repair" GO:0006281 and "cellular response to DNA damage stimulus" GO:0006974.*
- 3. Collecting GM co-expressed genes from humans only. (retrieved from public RNA seq data).*

The Reviewer correctly asserts our initial compiling of genome maintenance genes by GO terms. However, this first list of GM GO terms is not processed further. The subsequent two compiling steps (2 and 3) are individually retrieved i.e., the GM protein interactors list (2) as well as the list of GM co-expressed gene (3) are added to the list of GO terms list to yield a final list of total 5912 GM proteins. No phylogenetic analysis is performed at this stage. Hence, it is only the GO terms list of GM genes (1) that comprise gene/protein terms from different organisms other than human. Specifically, in the AMIGO database we find a total of 26,663 unique GM GO terms (DNA repair, DNA damage, Mitosis, Cytokinesis) across all species. Of these, our selected seven organisms GM GO terms represent: 284 unique E. coli GO terms (DNA repair GO: 0006281, DNA damage GO: 0006974, and cytokinesis GO: 000910), 413 unique S. pombe GO terms (DNA repair GO: 0006281, DNA damage GO: 0006974, cytokinesis GO: 000910, and mitosis GO: GO:0000278), 548 unique S. cerevisiae GO terms (DNA repair GO: 0006281, DNA damage GO: 0006974, cytokinesis GO: 000910, and mitosis GO: GO:0000278), 340 unique C. elegans GO terms (DNA repair GO: 0006281, DNA damage GO: 0006974, cytokinesis GO: 000910, and mitosis GO: GO:0000278), 462 unique D. melanogaster GO terms (DNA repair GO: 0006281, DNA damage GO: 0006974, cytokinesis GO: 000910, and mitosis GO: GO:0000278), 497 unique A. thaliana GO terms (DNA repair GO: 0006281, DNA damage GO: 0006974, cytokinesis GO: 000910, and mitosis GO: GO:0000278), and 1091 unique H. sapiens GO terms (DNA repair GO: 0006281, DNA damage GO: 0006974, cytokinesis GO: 000910, and mitosis GO: GO:0000278).

Hence, the GM GO terms list yields 3,635 unique terms (across the seven species), 390 GM protein interactors (humans only), and 2820 GM co-expressed genes (humans only) totaling 6845 terms.

What are the associated GO terms which are not related to your interest?

Our response

We appreciate the Reviewers concern about our retrieval of GO terms, and we hereby elaborate on this important method more clearly. In the three-step data compilation we have dealt with the GO terms outside of our focused interest accordingly:

- 1. GM GO terms compilation from the AMIGO database: All GO terms not included in the list of GM GO terms (DNA repair, DNA damage, Mitosis, Cytokinesis) AND not from the selected seven species are ignored.*
- 2. GM Interactome from the IID dataset and gene co-expression analysis: The compilation of GM protein interacting pairs and gene co-expressed pairs were performed in two sub steps:*
 - By only focusing on human proteins/genes that pair with proteins/genes retrieved from GO terms “DNA repair” GO:0006281 and “cellular response to DNA damage stimulus” GO:0006974.*
 - Of these pairs, only those protein/gene pairs where one protein/gene is NOT among the list of human GO terms comprised by “DNA repair” GO:0006281 and “cellular response to DNA damage stimulus” GO:0006974 were compiled. Hence, human protein/gene pairs where both proteins/genes are already established DNA repair factors were not included (due to being only ‘confirmatory’). In addition, in case of the PPI derived pairs, we focused on only protein interacting pairs that were identified in at least two independent proteomics studies. Hence, pairs that were identified in only one proteomic study were ignored. Thus, the final list of “putative” (hitherto unannotated) GM proteins that paired with an established DNA repair protein in at least two independent studies comprised 441 proteins. Subsequent filtering of this list of proteins in the CRAPome database yielded final 390 proteins that were then considered in our further analyses.*

2- Regarding the GO terms analyzed in the work. Terms were selected according to which category?

The GO: 00006281 is nowhere to be found in GO-related databases. For instance, a quick search gives five associated results <https://www.ebi.ac.uk/QuickGO/searchterms/GO:00006281>. Ideally, the results presented in the submitted work, should stand in the light of the current GO graph.

Our response

We are grateful to the reviewer for pointing this out. The GO terms selected for the survey are established terms as devised by the THE GENE ONTOLOGY RESOURCE. The GO term stated in the manuscript contains a typo. The correct term is “GO:0006281”, which captures DNA repair genes. We apologize for the error and for the confusion that it caused. We have corrected the error in the revised manuscript.

Perhaps an alternative way (is) to select also proteins is to extract the GO terms from the protein domains of the list, and extract those more specific for each category.

Our response

We thank the Reviewer for the suggestion. We initially wanted a starting list of GM proteins to be as comprehensive as possible to include as many protein domain structures as possible. Hence, this step in the pipeline represents a data compilation step rather than a refinement strategy, as explained above.

3- Regarding the PPI, there is a recently release dataset of PPI devoted to DDR, which I believe to be more adequate to use for the screening purposes of this paper, <https://doi.org/10.1016/j.inccels.2023.04.007> . Ideally, these should constitute a more curated dataset to be used for screening, as it provides assemblies.

Our response

We appreciate the Reviewer's suggestion. The mentioned study is interesting in that the authors used 21 (not endogenously) affinity tagged DDR proteins and subsequent affinity purification to establish 405 DDR interaction in humans of which 295 had not previously been annotated in public proteomics databases. However, while these new DDR candidates are intriguing, we also realize that historically high throughput interactome approaches have been plagued by varying degrees of high false discovery rate. Therefore, a thorough biochemical/functional validation of the candidates identified in this study will need to be performed. To minimize chances of dealing with false positives in our present work, we have ourselves intentionally focused solely on protein interactors that are recapitulated in at least two independent interactome proteomics studies. For that reason, we feel that the approach used in our present manuscript is safer.

4- The co-expression networks are by default extremely noisy (10.1186/s13059-019-1700-9). There are many reasons why genes can be co-expressed to together, and not being involved in a particular function. It is not clear how confounding effects have been removed from the “standard pipeline” from TCGA biolinks. There are more adequate methods, alternative to Spearman correlation, see for instance: 10.1093/bioinformatics/btab041.

Our response

We agree with the reviewer that confounding effects in gene co-expression networks represent a genuine concern to address. While we have had difficulties making the code work for the signed distance correlation as proposed in the study suggested by the reviewer, we have had more success with using Cemitool package that uses weighted gene correlation analysis (WGCNA) to derive networks of genes with correlated expression, which outperforms gene co-expression analysis using spearman correlation. The WGCNA approach deviates from the simple Pearson correlation coefficient (PCC) of linear relationships by using a multistep analysis of correlated genes including soft-thresholding, transformation to adjacency matrix, topological overlap matrix, and finally hierarchical clustering to ultimately identify modules or clusters of highly interconnected genes that are more unlikely to occur because of

confounding effects. We have assessed the new gene co-expression list including the new gene pairs as described in the Method section. The code and source data for this analysis will be provided at the provided link at the data availability section.

We also performed focused gene co-expression analysis of two selected candidate genes, namely the FAM72 family and MIAP genes. These analyses were performed using the Gene Expression Profiling Interactive Analysis 2021 (GEPIA2) resource based on deconvolution analysis of the normal samples from The Cancer Genome Atlas (TCGA) and Genotype-Tissue Expression (GTEx) databases. The resulting gene co-expression profiles are supported by recent experimental work implicating these proteins in the genome maintenance pathways doi: 10.1038/s41586-021-04093-y, doi: 10.1038/s41586-021-04144-4, and doi: 10.15252/embr.202255778. We have therefore kept these gene co-expression analyses in the revised manuscript.

5-The pipeline, the authors put a lot of effort to run many methods to achieve the same thing. For instance, they initially use HHblits and PSI_BLAST for remote homology. Since it is long known that PSI-BLAST provides a great deal of false positives (more than HHblits), I failed to grasp the relevance of running PSI-BLAST, unless they provide the relative contribution of each one for the identification of the domains, which is not given in the paper.

Our response

We agree with the reviewer that HHblits usually performs better than PSI_BLAST in remote homology searches. We have mostly used the HHblits method to produce MSAs to build the HMMs. There are practical differences between the HHblits and PSI-BLAST database availability in the HH-suite, however, which makes a direct comparison difficult. Using the HHblits the Uniref30_2023_02 database is the default option and using PSI-BLAST many databases are available including the non-redundant (nr) databases used in our survey. The Uniref30_2023_02 database typically produces much fewer matches due to the clustered representative sequences and therefore does not produce as comprehensive and diverse MSAs as the PSI-BLAST using the nr database. In rare borderline cases, however, our experience is that the MSAs build from re-iterative PSI-BLAST searches can build HMMs that obtain sequence matches with slightly higher significance than the corresponding HMMs build from the Uniref30_2023_02 database searches. One should of course be cautious with the conclusions drawn from such borderline cases and confirmation using reciprocal (candidate - > template) searches are necessary, as outlined above and performed in our present study. We have now in the revised manuscript for all identified domain candidates provided probability plots (cumulative histogram of probability values) of e-values.

6- They use Alpha Fold to predict the structures, in its multimer conformation. Some details are missing, for example, are the previously generated MSA fed into the pipeline (AF2 generates HHBLITS itself to run the evoblocks)? Are pdb templates also used? Those are relevant issues. Why is not Foldseek used systematically in the identified proteins?

Our response

We have used the Colabfold notebook that implements the MMseqs2 search engine to build MSAs used for profile-HMM searches and subsequently evo blocks. The protein complex

models were not generated with the help of PDB templates. We have added these important details to the method section in the revised manuscript, as suggested by the reviewer.

We agree that FoldSeek is a great software that could in principle be used for all searches instead of the hhsuite. FoldSeek exceeds other Fold recognition software such as DALI because it in addition to the fold recognition algorithm also integrates sequence conservation from profile-HMM searches. We have not used Foldseek for all the searches for two reasons. First, much of the analysis was performed before the release of Foldseek. Second, we find that FoldSeek is less sensitive in remote homology detection than the HHsuite (although Foldseek uses the HHblits and HHsearch executives to perform the profile HMM searches).

Line 237. All paralogues tend to keep the same structure, I do not grasp the relevance of the sentence. They could participate in the same pathway, doing different things.

Our response

We apologize for the confusion. What we meant was that the FAM72A-D paralogs' structures are predicted to be highly similar. Not just their similar structural folding, but identical structure down to the random coil structures between sheets and alpha-helices. This is rare.

line 422, where does the house keeping gene list come from?

Our response

We apologize for the lacking information. We have added the source of the house keeping gene list to the revised manuscript in the Data Availability section.

line 429, the github repository to evaluate the code is not accessible, so reliability and reproducibility of the code cannot be evaluated.

Our response

We apologize for the lacking information. The github source with the code has been added to the revised manuscript in the Data Availability section.

Line 453, IQtree is a better choice of making phylogenies from distantly related sequences, and provides real bootstrap values.

Our response

We thank the Reviewer for this suggestion. We have assessed the maximum-likelihood (ML) phylogenetic analyses in IQtree and found that it performed essentially the same as FastTree 2.1.

line 544, typo two dots.

Our response

We have corrected the typo in the revised manuscript.

line 553, A clarification: fold similarity does not mean necessarily to be evolutionary related nor to be functionally similar.

Our response

We are aware of this distinction. That is why structural fold comparisons alone by fold detection software such as DALI are poor at predicting evolutionary relationships. The AlphaFold2 3D structural modelling, however, is based on sequence conservation recognition. Whether the conserved signature motifs of such domains are actually lost during the profile-HMM-based MSA generation requires detailed experimental validation of each predicted domain. Our inspection of the literature and the hitherto published functions of the domains that we have predicted in GM proteins, as well as our own biochemical validation in this work, suggest that the AlphaFold2 predictions are reliable, as outlined above. In addition, we believe that illustrating the profile-HMM search results (used by AlphaFold2) using both forward and reciprocal probability plots provide a reliable validation tool to judge the redundancy of the remote homologies detected by the profile-HMM method on any given domain.

line 579, the sentence about Clustal is confusing, it reads as the MSA is generated using this method.

Our response

We apologize for the confusion and have clarified the sentence in the revised manuscript. (p.22, line 694).

Reviewer #2 (Remarks to the Author):

Exploring the structural landscape of DNA maintenance proteins
KB Schou et al.

The paper describes a systematic survey of DNA maintenance proteins from bacteria to humans using profile-to-profile models. The study aims to identify new DNA maintenance protein candidates and annotate new domains in established DNA maintenance proteins. The researchers collected annotated DNA damage response (DDR) and mitotic proteins from various species and used profile-to-profile search schemes to analyze the protein sequences. They applied iterative profile-HMM searches and validated the candidate domain structures using AlphaFold. Through their analysis, they identified several new conserved domains and expanded the understanding of known protein families.

Some of the key findings of the study include:

- Discovery of new DNA binding domain families: The analysis uncovered previously unknown members of DNA binding domain families, such as MYB/SANT, TUDOR, OB-fold, Mis18/yippee, SAP, WSD, and KU70/KU86 beta-barrel domains.
- Identification of new mitotic protein family members: The study identified new members of six protein families involved in mitotic functions, particularly kinetochore-related proteins.
- Annotation of new conserved domains in DNA repair proteins: The analysis revealed new conserved domains in DNA repair proteins, including the Integrator (Int) complex subunits and potential poly(ADP-ribose)-related enzymatic processes.
- Expansion of the understanding of DNA-binding domains: The study expanded the superfamily of MYB/SANT domain-containing proteins and predicted new SAP domains in various proteins, including PARP1.

Overall, the study highlights the use of computational methods to explore the structural landscape of DNA maintenance proteins and discover new domains and relationships. The findings provide valuable insights into the repertoire of genome stability caretakers and contribute to a better understanding of DNA maintenance processes.

The following comments are suggested to improve the paper.

1. The technique of Profile HMMs has existed for a long time. The authors should cite literature that supports their notion that GM proteins have not been subjected to systematic comparative analysis by computational strategies and some of the reasons as to why.

Our response

We appreciate this important comment: In the introduction we cite, to our knowledge, the only previous attempt to systematically survey the GM proteins across species by computational strategies (Aravind et al., 1999), which used the less sensitive PSI-BLAST method. One can view our method as an updated and advanced version of this previous computational survey, stimulated by this request from the reviewer. Furthermore, we have now in the revised manuscript also referred to other, more focused computational surveys that were performed using specific domains in question such as the OB fold and BRCT domains. Please, see the introduction on page 3 in the revised manuscript. In addition, we have justified our use of profile-HMMs in systematic remote homology searches at page 3, line 80 and further.

2. The work needs to mention prior progresses made in this field (GM proteins) using computational means (to provide a bit of background) and where this work picks up from and the approach used here to produce these findings (a high-level discussion).

Our response

We agree that to improve the manuscript, we should provide a better background for this survey. We have followed the Reviewer's request and added passages in the introduction summarizing previous computational efforts to study GM proteins. See page 3.

3. The excerpt lacks a clear and well-structured introduction. It would be beneficial to provide a more concise and organized overview of the research question, methods employed, and key findings. This will enhance the readability and comprehension of the paper for the intended audience.

Our response

We agree that the Introduction should better convey the theoretical and scientific background of the study. We have in the revised manuscript modified the Introduction to clarify the research question, methods employed, and key findings. See page 3. In addition, we have added an overview of the computational survey highlighting our research question followed by a short summary of our results from the analysis (see Result section page. 3 and 4).

4. Some recent techniques like DeepInsight (PMID: 31388036) and Transformer model (<https://arxiv.org/abs/1706.03762>) are used in extracting relevant features for deep learning and/or machine learning methods. It would be good to describe these techniques (DeepInsight and Transformer model) and explain why the profile-profile HMM model was selected.

Our response

We thank the Reviewer for these interesting suggestions. We didn't know about this software prior to the finalization of this study. Most of the analysis used in our survey was performed before the releases of the DeepInsight and Transformer model software. However, we will take into consideration these approaches in our future research. One caveat concerning DeepInside, however, is that protein sequences are strings of amino acids and the DeepInsight methodology would need to transform these sequences into a format resembling an image. Subsequently, this transformation should capture by means of the convolutional neural network meaningful patterns or features in the sequences. Such a procedure would probably require laborious efforts perhaps too convoluted and uncertain for remote homology detection and/or clustering of protein structures.

5. The excerpt mentions the use of profile-to-profile models and iterative profile-HMM searches, but it lacks specific details on these methods. Providing a more detailed description of the computational strategies employed would improve the reproducibility and clarity of the study. In particular, in lines 438 to 446, a reader would like to know what are the results/statistical bias in more detail, more description of coiled-coils and heptad repeat analysis. Providing details will be beneficial. Also in line 441 and 442, the best performance using those methods are not presented.

Our response

We agree that our initial reasoning for the profile-HMM method outlined in the manuscript could benefit from a more detailed description of the computational strategies used. We have followed the Reviewers suggestion and added further background and details to the Introduction and method section (see Introduction page 3 and Methods page 14-20).

Profile-HMM searches performed in the HHpred software produce besides match scores (E-values and probabilities) also scores for secondary structures and coiled-coil (i.e., heptad repeat) propensities along the query sequence length. Hence, in cases where proteins bear coiled-coil or low complexity (disordered) regions these were detected by HHpred in our initial searches. Coiled-coil regions are highly redundant across human proteins and therefore compromises specific homology searches, hence the statistical bias. We therefore performed a second search where such regions were deleted from the search query sequence. We have added this elaboration in the Method section page 16.

Regarding the assessment of the profile-HMM software, we have followed the Reviewers question and added a figure showing the initial performance test of various profile-HMM methods on three selected domains (Fig. 1c in the revised manuscript). The COACH software was not tested for this purpose as the software has been discontinued from the webserver doi: 10.1093/bioinformatics/bth091.

6. Validation and Experimental Results: The excerpt briefly mentions experimental validation of the predicted domains in SPIDR and FAM72 protein families. However, it does not provide specific details about the experimental methods used, the results obtained, or any statistical analyses performed. Including these details would strengthen the paper's scientific rigor and credibility.

Our response

We agree that the method section needs further experimental details on the biochemical validation of SPIDR and FAM72B protein families. We have thus in the revised manuscript extended elaborated our descriptions of the methodologies used and experimental details (see Methods page 18-20). These newly added important pieces of information include methods of proteomic, RNA interference, immunoblots, cloning, and reagents i.e., siRNA oligos, biotinylated DNA species etc. See Method section in the revised manuscript. The immunoblots are representative results as indicated in the figure legends (which is one of the modes of presentation acceptable by Nature family journals). Hence, no statistical analysis was performed for those experiments. Statistical analysis for computational profile-HMM searches and phylogenetic assessment are described in the following work doi: 10.1093/bioinformatics/bti125 and doi: 10.1186/s12859-018-2053-1.

7. Discussion and Interpretation: The excerpt briefly mentions the identification of new domain families and their potential implications but does not delve into a comprehensive discussion or interpretation of the findings. The authors should provide a more in-depth analysis of the functional implications of the newly identified domains and their potential roles in DNA maintenance and repair processes.

Our response

We agree that the Discussion section should better convey our interpretation of the findings and their implementations. In the revised manuscript we have discussed our discoveries further to the extent that the Nature Communication format limitations allow it.

8. Language and Writing Style: The excerpt contains several grammatical and typographical errors that need to be addressed. Additionally, the writing style can be improved to enhance clarity and coherence. Proofreading the manuscript thoroughly and revising the language to ensure a clear and concise expression of ideas would be beneficial. Some examples are:

a). Page 2, Line 40: typo “fully5”

b). Page 2, Line 61: Grammar “... identified in an multiple sequence ...”. ‘an’ should be ‘a’.

Our response

We apologize for the several grammatical and typographical errors and have corrected them accordingly. In addition, we will attempt to clarify the language to better convey our reasoning. The changes have been made throughout the revised manuscript.

Reviewer #3 (Remarks to the Author):

In this manuscript, Schou et al. report the identification of a large number of new human proteins involved in the DNA damage response (DDR) and genome integrity maintenance. The study is based on a sophisticated bioinformatics strategy, complemented by robust proteomics and some cellular biology and biochemistry of selected examples of the presented proteins, to validate this approach. The manuscript provides a significant advance and would likely be of interest to a broad audience of researchers in several fields. In my opinion, the most valuable parts of this study are twofold: a) The well explained innovative bioinformatics strategy that in itself should be also applicable to other fields of biomedicine and hence help discover many more proteins (related to other cellular functions) based on the principles applied here, and then b) The ‘catalogue’ of the new proteins (and domains) itself, the future detailed functional characterization of which will most likely be taken up by many research labs with interests in the individual protein families whose new members are presented in this manuscript. The work is within the scope of Nature comms, it is overall a good candidate for the journal, and its character makes it probable that it will in time be cited by other groups, thanks to its ‘discovery’ of new protein members of families implicated in important biological processes and pathologies.

Our response

We thank the Reviewer for the positive comments and have followed the Reviewer’s request to improve the manuscript along the lines suggested.

Nevertheless, there are several issues that in my opinion need strengthening to better support the conclusions and enhance the impact of this dataset before it can be fully recommended for publication.

Specific criticisms and suggestions:

1. As to the powerful bioinformatics strategy presented here, it could still be improved to some extent: Thus, while the HMM-based predictions of protein domains are accurate and paved the way for the revolution in artificial intelligence-based structural predictions of protein domains by e.g., AlphaFold, it is generally recommended to perform additional statistical assessment of the identified candidates. This additional support should help to rule out the possibility of false positive candidates, and it is especially critical for validation of the shorter homology regions.

Our response:

As discussed above for the Reviewer 1's requests, we have produced probability plots for a selection of candidates to validate the reliability of the remote homology searches. These plots show the e-values and Viterbi raw scores and give in all cases a robust indication that the candidates show genuine homology to the search queries.

2. It would be informative to include (perhaps in the Supplementary material and for at least some of the proteins or domains) multiple sequence alignments of candidate families to see in detail the conserved properties of the identified domains.

Our response:

We agree with the reviewer that adding additional multiple alignments of the new GM candidates including data from more organisms would be interesting to a broader audience of the GM research field. In the revised manuscript, we have followed the Reviewer's request and added examples of expanded alignments to the Supplementary Figures (Supplementary Fig. S8, S9, and S10).

3. All presented DDR candidates are of human origin. It would be informative for the broader research community to include new DDR protein candidates from other species e.g., from the model organisms used to collect the initial protein domain list (at least the most some attractive protein families covered by this work).

Our response:

We agree with the reviewer that adding additional new candidates from more organisms are also interesting for a broader audience of the GM research field. We are currently providing examples of the profile-HMM search outputs using queries from organisms other than humans, which will be deposited at the github repository <https://github.com/Schoulab/Genome-maintenance-structures>. The full list of profile-HMM searches will be available at the time of publication.

4. I do realize that the identified protein domains and candidate proteins are too numerous to require functional validation for all, or a really deep characterization. Nevertheless, some additional functional insights into some attractive protein, such as the FAM72B presented in the manuscript, would further strengthen the conclusions and relevance for biology. For example, what phosphor-RPA species is affected by FAM72B-depletion and in turn what downstream mediators/effectors within the DNA damage network are affected? Furthermore, are the FAM72B-depleted cells hypersensitive to DNA damage?

Our response:

We have followed the request of the Reviewer and performed additional experiments on FAM72B to provide more biological insights into its potential function in cells. Immunoblot analysis of the human U2OS cells depleted for FAM72B using FAM72B siRNA indicates that the extent of the commonly used marker: phosphorylation of RPA at serine residues 4/8 became apparent essentially to the parallel positive control scenario of exogenously triggered DNA damage compared with cells depleted of FAM72B while not exposed to exogenous DNA damaging insult. We thank the reviewer for this suggestion, as the result indicates a significant extent of endogenous DNA damage as a consequence of the depleted FAM72B protein (and hence its function). Furthermore, we also analyzed RPA1 phosphorylation on serine 33, a marker that is more specifically indicating some form of signaling response to replication stress, again consistent with our proposed candidacy of FAM72's role in genome maintenance. The reviewer also asked about potential downstream mediators in such response. To this end, we re-examined the phosphorylation status of the downstream Chk kinases Chk1/2, which appear to be largely unaffected by the FAM72B loss. Since these latter, now repeated, analyses of Chk1/2 essentially recapitulated the immunoblots presented in the original submission, we have kept the immunoblot data on Chk1/2 as presented in the first submission. Taken together, these combined results on RPA phosphorylations and Chk1/2 phosphorylations are intriguing, suggesting an unorthodox replication stress form that activates the main upstream kinases such as ATR (leading to RPA phosphorylations) yet largely avoiding the activation of the downstream kinase Chk1 that forms one of the downstream branches of the ATR-activated global response - in this case likely bypassing the claspin-mediated Chk1 activation route and possibly relying on ATR targeting of a set of direct ATR substrates (such as RPA1) yet without involving Chk1. The lack of Chk2 activation further indicates that the genotoxic stress triggered by FAM72 depletion doesn't seem to translate to immediate DNA double strand breaks, since otherwise the ATM-Chk2 axis would be expected to respond as well. Overall, while inspiring, these findings will benefit from a much more comprehensive future functional analysis to pinpoint exactly the consequences of FAM72B depletion and the cellular response to such stress. At the same time, the robust activation of the RPA phosphorylation signaling is consistent with the role of FAM72, identified in our survey, within the DNA integrity maintenance network. We are presently investigating the cellular outcomes and viability of knockout and mutations within the FAM72A-D genes. This entails both knocking out the genes and then complementing wildtype versions of FAM72A-D for accurate control. The findings from this analysis will be presented in a follow-up study on the role of FAM72A-D in the DNA damage response (DDR).

5. Another attractive finding of the manuscript is the predicted association between C1ORF146 and other ERCC4 domain proteins, but this part would benefit from further support by experimental validation, such as through selected example(s) of co-immunoprecipitation from suitable human cell model(s). Such additional experimental data

would further support the interaction predicted by the AlphaFold Multimer analysis shown in the manuscript.

Our response:

We thank the Reviewer for the positive comment and have followed the Reviewers request. It turns out that C1ORF146 is one of the genes with the most restricted mode of expression, essentially being specific only for the germ cells, both spermatocytes and oocytes, at a specific stage when these cells undergo meiotic divisions that requires specific genome maintenance events including dealing with crossing overs etc. (see ref. doi: 10.1126/sciadv.aau9780 for the identification of mammalian orthologs' role in meiosis). While this circumstance validates the role of C1ORF146 in one important aspect of genome maintenance, the strict tissue/cell-type restriction to a substage of spermatocyte and oocyte maturation prevented us from performing co-immunoprecipitations suggested by the reviewer, as obtaining human germ cells from donors was unfeasible on ethical grounds, and any analyses of e.g. murine germ cells were facing opposition by our internal ethical committee, since we would need to kill large numbers of animals in order to obtain quantities of minor subset of e.g. spermatocytes undergoing meiosis that would be sufficient to obtain enough protein for a series of repeated co-immunoprecipitations required to check the postulated protein(domain) interactions. Such cross-species analysis would very likely also face technical issues of antibody specificity, therefore this was the only analysis that we could not perform within the period of revision, due to factors that are largely beyond our control.

REVIEWERS' COMMENTS

Reviewer #1 (Remarks to the Author):

Dear authors,
Thank you for addressing and/or discussing all my comments and suggestions.
I have truly enjoyed your work.

I have only three points here:

1- Species names should be in italic font, everywhere.

2- line 537 points to an empty github repo, <https://github.com/Schoulab/Genome-maintenance-structures/>
please add the data here <https://github.com/ELELAB/DDR-candidates>, or in zenodo.

3- Just to mention, regarding the point 1 of reviewer #2, in addition to the classic Aravind's 1999, there is previous research using a similar conceptual computational approach in related proteins. In the 2014 paper (<https://academic.oup.com/mbe/article/31/4/940/1107429>) you can find the usage of profile-based analyses within a comparative framework done in several proteins pertinent to this work, also including phylogenetics analyses.

Reviewer #1 (Remarks on code availability):

I did not review the code of points I requested since the repo is empty.

Reviewer #2 (Remarks to the Author):

Exploring the structural landscape of DNA maintenance proteins (R1)
KB Schou et al.

Thank you for addressing my previous comments on your manuscript. I appreciate the efforts made to revise the paper, and I find the majority of your responses satisfactory. Based on this, my recommendation is for 'Minor Revisions'.

However, I still have concerns regarding Comment #4. While I understand that implementing the DeepInsight (<https://doi.org/10.1038/s41598-019-47765-6>) and Transformer models (<https://arxiv.org/abs/1706.03762>) may not have been feasible in the context of your study, I believe it would be beneficial to include a discussion of these techniques in the Introduction section, or where most appropriate. Explaining why the profile-profile HMM model was preferred over these alternatives would provide readers with a complete understanding of the choices made during your research.

Reviewer #2 (Remarks on code availability):

The package provided a README file, however, it can be improved further such as by adding some examples of how to execute and what to expect.

Reviewer #3 (Remarks to the Author):

The authors have thoroughly addressed my concerns and appear to have satisfactorily addressed the concerns of the other reviewers.

Response to Reviewers' comments to our revised manuscript NCOMMS-23-23369B:

Overall, we highly appreciate the comments of all Reviewers. In our Re-revised manuscript, we have: a) addressed all 3 remaining (minor/textual) issues raised by Reviewer 1 and the remaining suggestion for more introductory information raised by Reviewer no.2. Our responses (in italics) are listed below, along with references to the changes made in the manuscript.

Ad. Reviewer #1 (Remarks to the Author):

Dear authors,

Thank you for addressing and/or discussing all my comments and suggestions. I have truly enjoyed your work.

Our response

We are delighted to see that our response pleased the Reviewer and we are grateful for this positive overall comment.

I have only three points here:

- 1- Species names should be in italic font, everywhere.

Our response

We apologize for this error and have changed all species names to italic in the main text, method section, and in the figure legends in the revised manuscript.

- 2- line 537 points to an empty github repo, <https://github.com/Schoulab/Genome-maintenance-structures/> please add the data here <https://github.com/ELELAB/DDR-candidates>, or in zenodo.

Our response

We agree that the scripts were not added to the folder referred to. We apologize for this inconvenience and have now added the scripts to <https://github.com/ELELAB/DDR-candidates/tree/master/hh-suite>

- 3- Just to mention, regarding the point 1 of reviewer #2, in addition to the classic Aravind's 1999, there is previous research using a similar conceptual computational approach in related proteins. In the 2014 paper (<https://academic.oup.com/mbe/article/31/4/940/1107429>) you can

find the usage of profile-based analyses within a comparative framework done in several proteins pertinent to this work, also including phylogenetics analyses.

Our response

We thank the Reviewer for this note. We agree that it should be cited in our manuscript, and this has now been done: in the Introduction, line 56 in our Re-revised manuscript.

Reviewer #1 (Remarks on code availability):

I did not review the code of points I requested since the repo is empty.

Our response

We thank the Reviewer for this notification. However, we have added all relevant scripts to the address <https://github.com/ELELAB/DDR-candidates> directory freely available.

Reviewer #2 (Remarks to the Author):

Exploring the structural landscape of DNA maintenance proteins (R1)
KB Schou et al.

Thank you for addressing my previous comments on your manuscript. I appreciate the efforts made to revise the paper, and I find the majority of your responses satisfactory. Based on this, my recommendation is for 'Minor Revisions'.

Our response

We appreciate and are thankful for the Reviewer's positive response to our revisions. We have addressed the remaining minor comments in the Re-revised manuscript, as explained below.

However, I still have concerns regarding Comment #4. While I understand that implementing the DeepInsight (<https://doi.org/10.1038/s41598-019-47765-6>) and Transformer models (<https://arxiv.org/abs/1706.03762>) may not have been feasible in the context of your study, I believe it would be beneficial to include a discussion of these techniques in the Introduction section, or where most appropriate. Explaining why the profile-profile HMM model was preferred over these alternatives would provide readers with a complete understanding of the choices made during your research.

Our response

We thank the Reviewer for the comment regarding his earlier comment #4, which is also relevant considering the recent developments in AI-based protein structural predictions. The recent release of AlphaFold3 (AF3) (<https://doi.org/10.1038/s41586-024-07487-w>) models are highly relevant to the Reviewer's comment, who emphasizes (among other points) the "convolution neural network" (CNN) method (used by the DeepInside software) relevant to

the latest attempts of computational protein-protein and protein-ligand structural predictions. The published study of AF3 addresses the previous deep learning attempts (including the graph neural networks (GNN) and CNN)) and provides evidence for AF3's better performance with respect to protein-ligand and protein-nuclei acid interactions. We have decided to address the Reviewer's comment by providing a newly predicted model of the SPIDR OB3 domain in complex with DNA (see new Figure 3k in the revised manuscript figures) using AF3, thus demonstrating that we have taken the reviewer's suggestion into account and have improved the predictive power of our study accordingly. In addition, we have also added a section in the introduction briefly discussing the advances of using the AF3 method over recent other deep learning methods.

Reviewer #2 (Remarks on code availability):

The package provided a README file, however, it can be improved further such as by adding some examples of how to execute and what to expect.

Our response

We thank the Reviewer for this notification. However, examples of the scripts for each of the computational procedures used in the data processing were already added to the <https://github.com/ELELAB/DDR-candidates> directory, which includes scripts (in Python format) as well as output example files (in csv format).

Reviewer #3 (Remarks to the Author):

The authors have thoroughly addressed my concerns and appear to have satisfactorily addressed the concerns of the other reviewers.

Our response

We thank the Reviewer for his/her positive comment.